# Epigenomic analysis of formalin-fixed paraffin-embedded samples by CUT&Tag

Steven Henikoff [1,2] ✉, Jorja G. Henikoff [1], Kami Ahmad [1],
Ronald M. Paranal [3], Derek H. Janssens [1], Zachary R. Russell [3],
Frank Szulzewsky [3], Sita Kugel[3] & Eric C. Holland [3]

For more than a century, formalin-fixed paraffin-embedded (FFPE) sample preparation has been the preferred method for long-term preservation of biological material. However, the use of FFPE samples for epigenomic studies has been difficult because of chromatin damage from long exposure to high concentrations of formaldehyde. Previously, we introduced Cleavage Under Targeted Accessible Chromatin (CUTAC), an antibody-targeted chromatin accessibility mapping protocol based on CUT&Tag. Here we show that simple modifications of our CUTAC protocol either in single tubes or directly on slides produce high-resolution maps of paused RNA Polymerase II at enhancers and promoters using FFPE samples. We find that transcriptional regulatory element differences produced by FFPE-CUTAC distinguish between mouse brain tumors and identify and map regulatory element markers with high confidence and precision, including microRNAs not detectable by RNA-seq. Our simple workflows make possible affordable epigenomic profiling of archived biological samples for biomarker identification, clinical applications and retrospective studies.

The standard workflow of surgical specimens is from the operating room into formalin (~4% formaldehyde) for a few days and then embedding into paraffin, cut into sections for histological analysis and stored as paraffin blocks. Even after long-term storage, formalin-fixed paraffin-embedded (FFPE) sections can be resurrected for application of modern sequencing-based genomic methodologies in ongoing and retrospective studies[1]. FFPE sample preservation has been in use for over a century, with billions of cell blocks accumulated thus far, and no end in sight[2]. Most genomic studies using FFPE samples have applied whole genome sequencing to identify mutations and aneuploidies, or whole exome sequencing to identify tissue-specific differences. However, chromatin profiling has the potential of identifying causal regulatory element changes that drive disease. The prospect of applying chromatin profiling to distinguish regulatory element changes is especially attractive for translational cancer research, insofar as misregulation of promoters and enhancers in cancer can provide diagnostic information and may be targeted for therapy[3]. However, there

has been limited progress in applying chromatin profiling techniques to FFPEs[4]. Although several methods have been developed for chromatin immunoprecipitation with sequencing (ChIP-seq) using FFPEs[5–10], ChIP-seq is not well-suited for small amounts of material that are typically available from patient samples. Furthermore, solubilization of such heavily cross-linked material is extremely challenging, requiring strong ionic detergents and/or proteases in addition to controlled sonication or Micrococcal Nuclease (MNase) digestion treatments.

Alternatives to ChIP-seq for chromatin profiling include ATAC-seq[11], DNase-seq[12], NicE-seq[13], FAIRE[14,15] and enzyme-tethering methods such as CUT&RUN[16] and CUT&Tag[17]. Modifications to the standard ATAC-seq protocol were required to make it suitable for FFPEs, including nuclei isolation following enzymatic tissue disruption and in vitro transcription with T7 RNA polymerase[18,19]. The same group also similarly modified CUT&Tag and included an epitope retrieval step using ionic detergents and elevated temperatures, which they termed

[1]Basic Science Division, Fred Hutchinson Cancer Center, Seattle, WA, USA. [2]Howard Hughes Medical Institute, Chevy Chase, MD, USA. [3]Human Biology Division, Fred Hutchinson Cancer Center, Seattle, WA, USA. ✉e-mail: steveh@fredhutch.org

FFPE tissue with Antibody-guided Chromatin Tagmentation with sequencing (FACT-seq)[20,21]. However, FACT-seq is a 5-day protocol even before sequencing, and the many extra steps required relative to CUT&Tag have raised concerns about experimental variability[4].

In this work, we wondered whether a fundamentally different approach to what has been described for FFPE-ATAC and FACT-seq might overcome the obstacles that have thus far been encountered in chromatin profiling of FFPEs. Rather than enzymatically breaking down the tissue for nuclei isolation, we use only heat and minimal shearing of the FFPE specimen, then follow our standard CUT&Tag-direct protocol with modifications. These include applying our Cleavage Under Targeted Accessible Chromatin (CUTAC) strategy, which preferentially yields <120-bp fragments released by antibody-targeted paused RNA Polymerase II (RNAPII)[22,23]. Because of the small size of the fragments released with CUTAC, it is relatively robust to the serious DNA degradation that occurs during cross-link reversal[24], and by attaching to magnetic beads and following the single-tube CUT&Tag-direct protocol, or by performing incubations directly on the slide, we minimize experimental variation. The resulting FFPE-CUTAC profiles could be used to confidently distinguish different mouse brain tumors from one another and from normal brain tissue, identifying potentially key regulatory elements involved in cancer progression.

## Results

### CUT&Tag streamlined protocol for whole cells

We originally introduced CUT&Tag with DNA purification by addition of SDS/Proteinase K followed by either phenol-chloroform-isoamyl alcohol extraction and ethanol precipitation or SPRI bead binding and elution for PCR[17]. Later we streamlined the protocol so that it could be performed in single PCR tubes using a 58 °C incubation in 0.1% SDS followed by excess Triton-X100, which sequesters the SDS in micelles, allowing efficient PCR[22]. However, this CUT&Tag-direct method was only suitable for up to ~50,000 nuclei, as more material was found to inhibit the PCR. To make CUT&Tag-direct applicable to whole cells, we have included 0.05% Triton-X100 in all buffers from antibody addition through tagmentation, which maintains cells permeable without disrupting nuclei and improves bead behavior. We have also increased the concentration of SDS and included thermolabile Proteinase K in the fragment release buffer. After digestion at 37 °C and inactivation at 58 °C, the SDS is quenched with excess Triton-X100 and the material is subjected to PCR, resulting in high yields with 30,000-60,000 whole cells (Supplementary Fig. 1a).

When applied to the H3K4me3 promoter mark, this modified CUT&Tag-direct protocol for native whole cells resulted in representative profiles that match those of native or fixed nuclei using either the original organic extraction method or CUT&Tag-direct (Fig. 1a). Based on MACS2 peak-calling and Fraction of Reads in Peaks (FRiP), we obtained slightly more peaks called and similar FRiP values for up to at least 100,000 native whole cells using the modified protocol (Fig. 1b, c), obviating the need to purify nuclei for CUT&Tag-direct[25] and AutoCUT&Tag[26].

### Temperature-dependent permeabilization of FFPE sections for CUTAC

To evaluate the ability of our approach to discriminate between archived samples, we chose paraffin blocks of three related mouse CNS tumor types, driven by distinct mechanisms. We compared 10-micron sections from FFPE blocks of tyrosine kinase active PDGFB-driven gliomas[27], ZFTA-RELA gene fusion-driven ependymomas[28], and YAP1-FAM118b gene fusion-driven ependymomas[29] to one another and to FFPE blocks of normal mouse brain.

The difficulty of performing CUT&Tag-direct on FFPEs is exacerbated not only by the severe chromatin damage caused by heavy formalin fixation but also by the large amount of cross-linked intra- and inter-cellular material that cells are embedded in. Both the FFPE-ATAC and FACT-seq methods require lengthy digestion with collagenases and hyaluronidase followed by 27-gauge needle extraction and straining liberated nuclei for processing. We reasoned that harsh treatments might not be necessary if the cells can be permeabilized sufficiently, and we were encouraged to attempt this approach by the fact that deparaffinized 5- to 10-micron FFPE samples on slides are routinely permeabilized for cytological staining with antibodies[1]. Also, there has been recent progress in preventing the most severe DNA damage to FFPEs by careful attention to buffer and heating conditions[24]. Accordingly, we performed manual shearing of deparaffinized 10-micron FFPE sections from tumor and normal mouse brains by dicing and scraping the tissue off slides with a razor blade followed by forcing the solution twenty times through a 22-gauge needle. We found that the Concanavalin A (ConA) beads used for standard CUT&Tag, bound sufficiently well to sheared FFPE fragments. This meant that all steps from antibody addition through to PCR could be performed on FFPEs following the same CUT&Tag-direct protocol used for nuclei and whole cells. In addition, the toughness of FFPE shards allowed

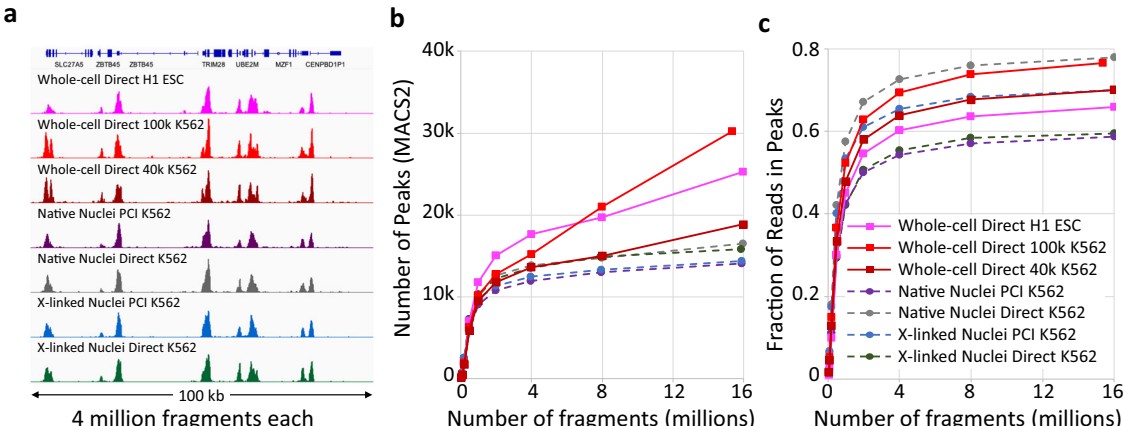

**Fig. 1 | High data quality from CUT&Tag-direct for whole cells. a** A comparison of H3K4me3 CUT&Tag tracks for human H1 (track 1) and K562 cells (tracks 2–7) at a representative 100-kb region of housekeeping genes. Group-autoscaled profiles for 4 million mapped fragments from each sample are shown. For Whole-cell Direct K562 samples either 100,000 (red) or 40,000 (brown) cells were used. **b**, **c** Graphs of Number of Peaks (left) and Fraction of Reads in Peaks (FRiP, right) and color-coded as in (**a**). Random samples of mapped fragments were drawn, mitochondrial reads were removed and MACS2 was used to call peaks using the narrow peak option. The number of peaks called for each sample is a measure of sensitivity, and FRiP is a measure of specificity calculated for each sampling from 50,000 to 16 million fragments. Nuclei data are from a previously described experiment[69]. Source data are provided as a Source Data file.

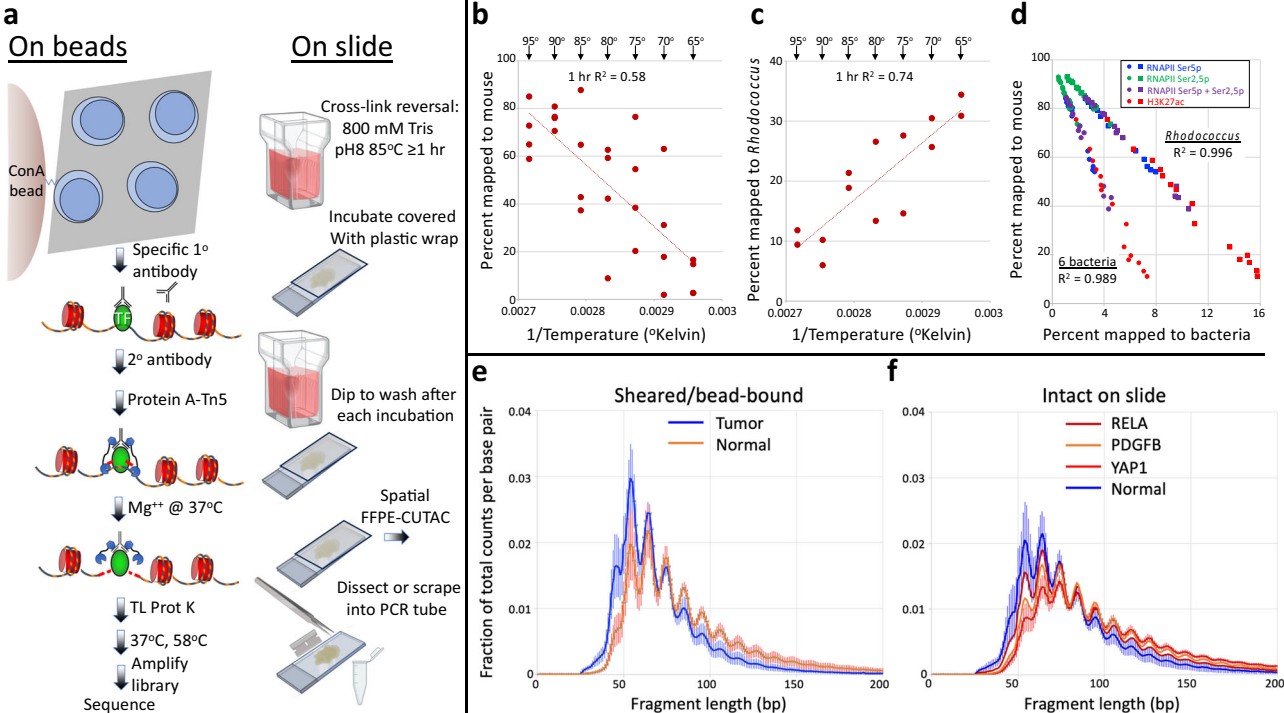

**Fig. 2 | High temperatures improve yield of small mouse fragments with FFPE-CUTAC. a** Scheme, where TL Prot K is Thermolabile Proteinase K (New England Biolabs). Created with BioRender.com. **b** Arrhenius plot showing the recovery of fragments mapping to the Mm10 build of the mouse genome as a function of temperature. Deparaffinized FFPEs were scraped into cross-link reversal buffer (20) containing 0.05% Triton-X100, needle-extracted, and divided into PCR tubes for incubation in a thermocycler at the indicated temperatures. **c** Same as (**b**) except for fragments mapping to the *Rhodococcus erythropolis* genome. **d** Scatter plots and $R^2$ correlations between total fragments recovered versus *R. erythropolis* and the summed totals for 6 other bacterial species discovered in BLASTN searches of unmapped reads (*Escherichia coli*, *Leifsonia* species, *Deinococcus aestuarii*, *Mycobacterium syngnathidarum*, *Vibrio vulnificus*, and *Bacillus pumilus*). **e** Comparison of average overall length distributions between tumor and normal brain, combining samples from all 3 brain tumors (YAP1, PDGFB and RELA). RNAPII-Ser5p: 15 samples; RNAPII-Ser2,5p: 15 samples; H3K27ac: 15 samples; 50:50 mixture of RNAPII-Ser5 and RNAPII-Ser2,5p: 14 samples. For each sample, mouse fragment lengths were divided by the total number of fragments before averaging. Lengths are plotted at single base-pair resolution. **f** Average length distributions for on-slide samples grouped by cancer driver transgene (YAP1: 12 samples; PDGFB: 7 samples; RELA: 12 samples) and Normal brain: 10 samples. Data are presented as mean values +/- SD in (**e**, **f**). Source data are provided as a Source Data file.

for hard vortexing and centrifugation steps that would have resulted in lysis of ConA bead-bound cells or nuclei.

Formaldehyde cross-links are reversed by incubation at elevated temperatures. Typical ChIP-seq, CUT&RUN and CUT&Tag protocols recommend cross-link reversal at 65 °C overnight in the presence of Proteinase K and SDS to simultaneously reverse cross-links and deproteinize. However, the much more extreme formaldehyde treatments that are used in preparing FFPEs have required incubation temperatures as high as 90 °C for isolation of PCR-amplifiable DNA for whole-genome sequencing[24,30,31]. High temperatures also contribute to epitope retrieval for ChIP-seq[5–10] and FACT-seq[20], and for cytological staining one protocol calls for epitope retrieval at 125 °C at 25 psi in a pressure cooker[32]. To optimize the temperature of incubation for DNA recovery and epitope retrieval for CUTAC on FFPE samples from mouse brain tumors, we incubated sheared FFPEs at temperatures ranging from 65 °C to 95 °C before ConA bead and antibody additions. We performed modified CUT&Tag-direct using low-salt tagmentation (CUTAC) with RNAPII-Ser5p and/or RNAPII-Ser2,5p and H3K27ac antibodies. Upon DNA sequencing, the fraction of fragments that mapped to the mouse genome showed a strong temperature dependence, where the highest temperatures (90-95 °C) showed the highest fraction mapping to the mouse genome (75%), and the lowest temperatures (65-70 °C) showed the lowest fraction (13%) (Fig. 2b). A relationship between cross-link reversal and incubation temperature has been determined to follow the Arrhenius equation[33]. As temperature dependence of mouse tagmented fragment recovery also followed the Arrhenius equation, cross-link reversal may be limiting for DNA fragment recovery.

## High temperatures preferentially reduce tagmentation of contaminating bacterial DNA

We were curious as to the identity of fragments generated by FFPE-CUTAC that did not map to the mouse genome. Using BLASTN against nucleotide sequences in Genbank it became apparent that there was a single species that consistently rose to the top of the list for all samples, the gram-positive bacterium *Rhodococcus erythropolis*. Mapping fragments to the *R. erythropolis* genome, we found that the entire genome was represented as expected if this species is a major contaminant of the mouse brain FFPEs in our study. Consistent with this interpretation, we found a high-temperature dependence of fragment release opposite that for mouse (Fig. 2c), consistent with *Rhodococcus* fragments competing with mouse fragments in the PCR. We also found a near-perfect anti-correlation between the fraction of fragments mapped to mouse and the fraction mapped to the *R. erythropolis* genome ($R^2 = 0.996$, $n = 59$) across all antibodies (Fig. 2d), with *Rhodococcus* accounting for 1-15% of the total fragments. As bacterial DNA is not chromatinized, it is unlikely to be protected from melting as well as mouse DNA, and so would not serve as a substrate for Tn5 tagmentation, which could account for the reduction in *Rhodococcus* contamination with increasing temperature.

To obtain a broader representation of species contaminating our FFPEs, we performed BLASTN searches of the RefGene Genome

Database using a sample of 300 multiply represented 50-bp reads not aligning to the Mm10 build of the mouse genome. A search of the bacterial genome subset returned hits to 208 species for ~2/3$^{rd}$ of the fragments, which implies that most of the unmapped reads were bacterial in origin. Although no other bacterial species were nearly as abundant as *R. erythropolis*, summing the fragment counts mapped to the six most frequently represented other species accounted for ~0.5–7% of the fragments and showed similar near-perfect anti-correlations to mouse ($R^2$ = 0.989, Fig. 2d). Efficiency was highest for RNAPII Ser2,5p (85% mouse, 2.5% *Rhodococcus*) and lowest for H3K27ac (38% mouse, 11% *Rhodococcus*). The lower efficiency of the histone modification than the RNAPII modifications might be attributed to susceptibility of lysine-rich histone tails to formaldehyde adduct and cross-linking damage, in contrast to the 52-copy lysine-free YSPTSPS heptamer comprising the C-terminal domain of Rpb1. Efficiency for FFPEs was also low for other histone modifications, resulting in poor signal-to-noise for the repressive H3K27me3 mark (Supplementary Fig. 2) and complete failures for H3K4me3 and H3K4me2, which were used with the original CUTAC protocol[22]. In contrast, histone H3K27ac, a mark of active enhancers and promoters, resulted in high mappability (Fig. 2d), perhaps because unlike H3K4 and H3K27 methylations, H3K27ac is not known to be bound by "reader" proteins, which may cause epitope masking when cross-linked to their histone tail substrates.

What is the source of *Rhodococcus* and other bacterial contaminants in our FFPEs, which derive from multiple FFPE sample preparations over a 2-year span? *R. erythropolis* isolates have been found to use paraffin wax as their sole carbon source, forming thick biofilms[34]. The species has also been proposed as an industrial biodegrader for removing the paraffin wax that remains on the inner surfaces of oil tanker holds after they are emptied[35]. We infer that most of the DNA fragments that do not map to mouse are derived from the paraffin used in embedding, with an advantage during PCR over the tissue derived DNA in not having been subjected to formalin treatment. We interpret the near-perfect anti-correlations seen for these genomes in different samples as reflecting a very uniform distribution of contamination for slides prepared at different times.

## FFPE-CUTAC performed directly on the slide

Although high temperatures and stringent washes were unable to completely eliminate bacterial contamination, we suspected that concanavalin A on the beads might have captured residual dead bacterial cells. To test this possibility we substituted amine-coated paramagnetic beads for ConA beads and found that when followed by a centrifugation pulse at 3000 g before magnetizing, amine-coated bound sufficiently well to sheared FFPE fragments that we obtained similar recoveries as with ConA beads. We also tested hot-aqueous deparaffinization by placing FFPE slides in a slide holder filled with cross-link reversal buffer and incubating overnight at 85 °C (Fig. 2a), based on previous reports showing that hot water suffices to melt and float off paraffin without the need for organic chemical pretreatment[36–39]. Finally, we tested a bead-free approach, in which overnight incubation at 85 °C was followed by incubations directly on the slide covered by plastic film and washes by immersion. We observed that either using magenetic beads without ConA or performing FFPE-CUTAC directly on the slide resulted in 99% mappability, completely eliminating residual bacterial contamination (Supplementary Data 1).

## Subnucleosomal fragment sizes from FFPE-CUTAC samples

Capillary gel profiles of FFPE-CUTAC libraries revealed insert sizes averaging ~60 bp (Supplementary Fig. 1b), despite inclusion of a 1-minute 72 °C PCR extension step in each PCR cycle intended to capture larger fragments from degraded template DNA. After DNA sequencing, we observed subnucleosomal length distributions

showing 10-bp periodicities typical of CUT&Tag peaking at ~60 bp for all antibody series (Supplementary Fig. 3a). By separately plotting the fragment length distributions for tumors and normal brains, we observed a conspicuous difference, where the length distribution was shifted with more longer fragments in tumor (median = 76 bp) relative to normal brain tissue (median = 65 bp) (Fig. 2e and Supplementary Fig. 3b). Fragment length distributions of RNAPII-Ser5p FFPE-CUTAC data using the on-slide protocol confirmed that the tumors yielded longer fragments than normal brain, with YAP1-FAM118b gene fusion-driven ependymomas showing the largest length increase relative to normal brain (Fig. 2f). In contrast, the two overall length distributions of *Rhodococcus* DNA fragments from the same tumor and normal samples closely superimposed (Supplementary Fig. 3b). This average shift to a longer fragment distribution for tumors is also seen for mitochondrial DNA from the same samples when compared to either normal brain or CUT&Tag mitochondrial DNA profiles from native 3T3 fibroblasts (Supplementary Fig. 3c). However, a small difference in the opposite direction was observed between liver tumor (median = 63 bp) and normal (median = 68 bp) FFPEs (Supplementary Fig. 3d), which suggests that the length differences seen between tumor and normal mouse brain are tumor-specific. Interestingly, both *Rhodococcus* and mouse mitochondrial fragments from FFPEs displayed a much weaker 10-bp periodicity relative to mouse brain FFPE nuclear and unfixed mouse mitochondrial fragments, respectively (Supplementary Fig. 3c), suggesting that the reduction in periodicity seen for DNA unimpeded by nucleosomes (bacterial and mitochondrial) is the result of DNA damage caused by fixation and cross-link reversal. The strong periodicity seen for mouse CUTAC profiles relative to non-chromatinized DNA of bacteria and mitochondria in the same samples might reflect partial protection from unreversed formadehyde fixation damage by RNAPII and other chromatin regulatory complexes characteristic of open chromatin[40].

## FFPE-CUTAC produces high-quality maps of active chromatin

To evaluate the accuracy and data quality of FFPE-CUTAC applied to mouse brain tumors, we compared tracks between FFPE-CUTAC and FACT-seq or standard CUT&Tag from the same study[20] using the same H3K27ac antibody (Abcam cat. no. 4729). Because of differences in cell types, brain tumors in our study and kidney or liver in the FACT-seq study, we limited comparisons of tracks to housekeeping genes that are expected to be similarly expressed in all cell types. Based on visual inspection of tracks from representative regions of the mouse genome, it is evident that H3K27ac CUTAC profiles show much cleaner profiles than those obtained using FACT-seq, with higher sensitivity than the data obtained for CUT&Tag controls of frozen mouse kidney (Fig. 3a–d). Likewise, clean profiles were also seen for RNAPII-Ser2,5p FFPE-CUTAC, where RNAPII-Ser2 phosphate marks elongating and RNAPII-Ser5 phosphate marks paused RNAPII.

For a systematic analysis of data quality, we called peaks using MACS2[41] and compared the number of peaks called and FRiP values. Both H3K27ac and RNAPII-Ser2,5p FFPE-CUTAC on RELA- and PDGFB-driven brain tumors showed much better sensitivity based on number of peaks called and much higher FRiP values than either H3K27ac CUT&Tag on frozen kidney or FACT-seq on FFPEs (Fig. 3e, f).

To determine the degree to which FFPE-CUTAC profiles capture regulatory elements, we took advantage of the Candidate *cis*-Regulatory Elements (cCRE) database generated by the ENCODE project, which called putative regulatory elements from all tissue types profiled. We used the 343,731 elements in the cCRE mouse database based mostly on DNAseI-seq, but also H3K4me3 and CTCF ChIP-seq. This resource provides a comprehensive standard for FFPE-CUTAC performance, insofar as CUTAC profiles correspond closely to both ATAC-seq and DNAseI-seq profiles[22]. For each dataset we rank-ordered cCREs based on normalized counts spanned by each element, which we plotted as a log-log cumulative curve, where a higher curve indicates

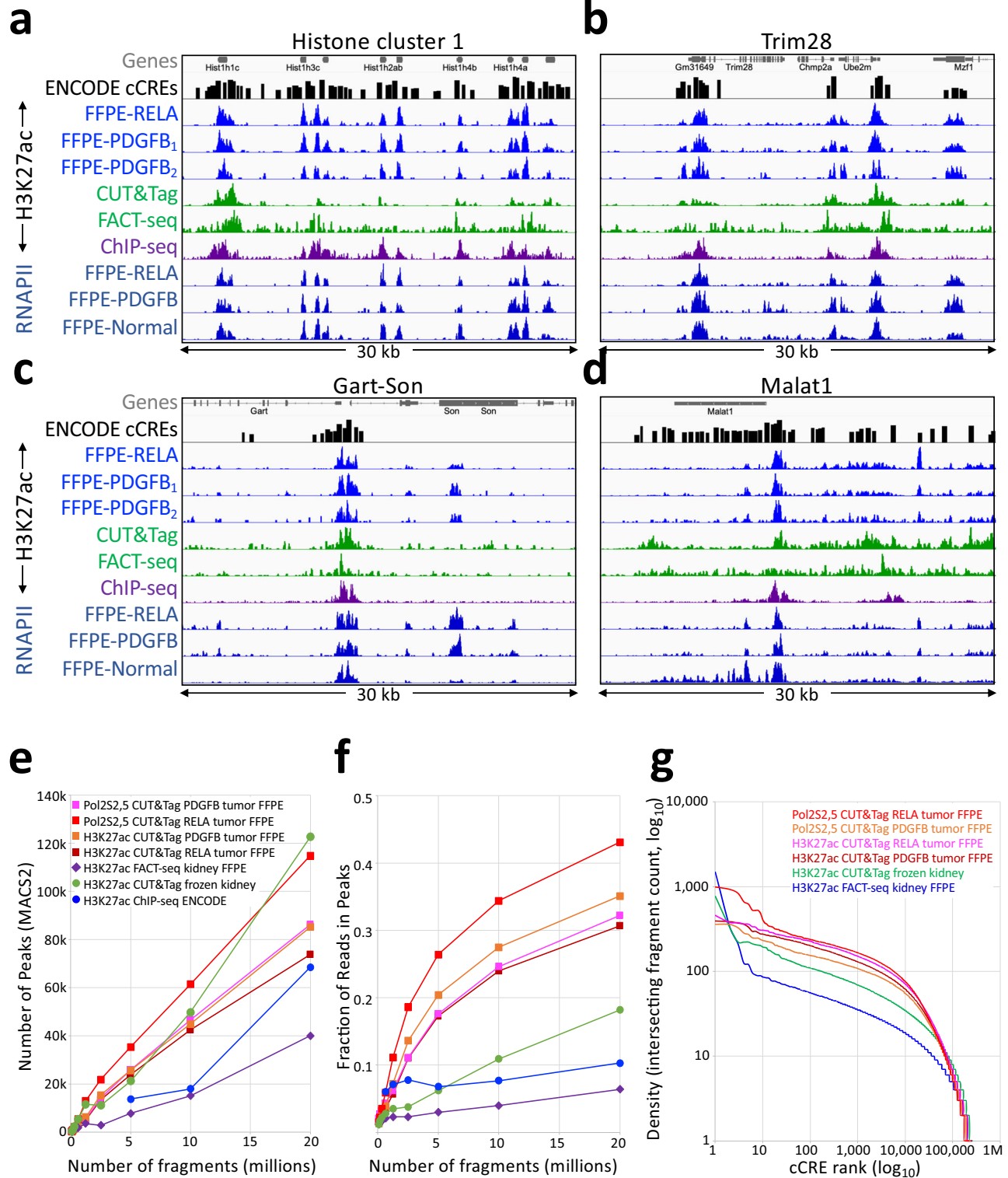

**Fig. 3 | Comparison of H3K27ac FFPE-CUTAC to FACT-seq and CUT&Tag of frozen unfixed samples. a–d** IGV tracks showing representative examples of housekeeping gene regions were chosen to minimize the effect of cell-type differences between FFPE-CUTAC (three brain tumors) and published FACT-seq and control CUT&Tag data (kidney). Forebrain H3K27ac ChIP-seq and ATAC-seq samples from the ENCODE project are shown for comparison, using the same number of fragments (20 million) for each sample. Also shown are tracks from FFPE-CUTAC samples using an antibody to RNAPII-Ser2,5p. A track for Candidate *cis*-Regulatory Elements (cCREs) from the ENCODE project is shown above the data tracks, which are autoscaled for clarity. **e, f** Number of peaks and Fraction of Reads in Peaks (FRiP) called using MACS2 on samples containing the indicated number of cells. **g** Cumulative $\log_{10}$ plots of normalized counts from 10 million mapped fragments intersecting cCREs versus $\log_{10}$ rank. Source data are provided as a Source Data file.

better performance in distinguishing annotated sites from background. By this benchmark, both H3K27ac and RNAPII-Ser2,5p FFPE-CUTAC brain datasets outperformed both FACT-seq on FFPEs and CUT&Tag on unfixed frozen kidney (Fig. 3g). We conclude that our FFPE-CUTAC protocol provides high quality data, even when compared to standard CUT&Tag.

## FFPE-CUTAC profiles distinguish brain tumors and reveal global upregulation

Nearly all strong peaks seen for H3K27ac and RNAPII-Ser2,5p FFPE-CUTAC corresponded to putative regulatory elements from the cCRE database, with concordance between FFPE-CUTAC, FACT-seq and ChIP-seq (Fig. 3a–d). To identify tumor-specific candidate regulatory elements we performed pairwise comparisons between three different mouse brain tumors (YAP1-, PDGFB- and RELA-driven tumors) and normal mouse brains. For each of the 343,731 cCREs we averaged the normalized counts spanned by the cCRE and performed pairwise comparisons over all cCREs with Voom/Limma[42], an Empirical Bayes algorithm, which uses the other datasets as pseudo-replicates to increase statistical confidence. We applied this approach to datasets from multiple FFPE-CUTAC experiments using antibodies against RNAPII-Ser5p, RNAPII-Ser2,5p and H3K27ac. We observed far more significant differences for comparisons between tumors and normal brains than between tumors, with more increases than decreases in tumors relative to normal brains (Fig. 4a–c and Supplementary Data 2a-d). For example, using RNAPII-Ser5p, there were 10,321 cCREs that differed between YAP1 and normal brain, 518 between PDGFB and normal brain, and 190 between RELA and normal brain at a False Discovery Rate (FDR) = 0.05, but only 10-63 cCREs that differed in pairwise comparisons between the three tumors (Fig. 4a and Supplementary Data 2a). Compared to normal brain, 92-99% of the differences were increases in the tumors. Approximately similar results were obtained using RNAPII-Ser5p (Fig. 4b and Supplementary Data 2b). For H3K27ac, the number of cCREs that increased was more extreme, with nearly half of the 343,371 cCREs significantly increased at the FDR = 0.05 level (Fig. 4c and Supplementary Data 2d). These results demonstrate that FFPE-CUTAC using antibodies against RNAPII or H3K27 marks distinguishes between the tumors and the normal brain samples with nearly all significant differences representing increases for the three tumors over normal brain.

As FFPE-CUTAC data quality is very similar between RNAPII-Ser2,5p and H3K27ac (Fig. 3), we attribute the conspicuous sensitivity differences in pairwise comparisons (Fig. 4a–c and Supplementary Data 2a–c) in part to the larger number of H3K27ac samples that Voom/Limma used for pseudo-replicates in calculating FDR. To balance the contribution of samples from each genotype, we merged datasets from multiple FFPE-CUTAC experiments for each antibody (RNAPII-Ser5p, RNAPII-Ser2,5p or H3K27ac) or antibody combination (RNAPII-Ser5p + RNAPII-Ser2,5p), then down-sampled to the same number of mapped fragments for each genotype. The three tumor and one normal genotype, each represented by four different antibodies or antibody combination, were compared pairwise with Voom/Limma. We observed the most differences between RELA and Normal (1,657) and between RELA and PDGFB (607) and the fewest differences between PDGFB and YAP1 (17) (Fig. 4d and Supplementary Data 2e). We conclude that FFPE-CUTAC can distinguish tumors from one another and from normal brains based on differences in cCRE occupancy of active RNAPII and H3K27ac marks.

## Increases in paused RNAPII pinpoint regulatory element differences

To identify gene regulatory elements genome-wide that best distinguish tumor from normal and between tumors, we performed Voom/Limma analysis using the maximum normalized count within each cCRE, rather than the average of normalized counts over the entire

cCRE. The most significant difference among all RNAPII-Ser5p cCRE comparisons is a sharp peak in a coding exon of the PDGFB gene, which is present in the PDGFB-driven tumors but absent in the normal brain (FDR = $5 \times 10^{-5}$, Fig. 5a). This example serves as an internal control, as it corresponds to the virally expressed PDGF-beta growth factor coding region that drives the tumor, even though this sample contained both normal brain and tumorous tissue. The other most significant and highly expressed differences between tumors and normal brain identify loci that have been reported as implicated in tumor progression. Among these are the SET domain-containing 5 (*Setd5*) promoter (Fig. 5b)[43], the Phosphoglucokinase (*Pgk1*) promoter (Fig. 5c)[44], which are also from the PDGFB-driven tumor and normal comparison, displaying clear differences between the tumors. Additionally, the cCREs in these genes show high signal in the RELA-driven tumor and low signal in the YAP1-driven tumor. Even more striking differences are seen for the next two most significant differences at the bidirectional promoter of the Insulin growth factor 2 (*Igf2*) (Fig. 5e) and the Collagen type 1 alpha 1 (*Col1a1*) gene promoter (Fig. 5d)[45,46], where the RELA-driven tumor shows a strong signal but there is no perceptible signal in the region for normal, PDGFB-driven and YAP1-driven samples. Conspicuous tumor-specific differences are also seen for four of the five cCREs with the highest signals with FDR < 0.05, including an intronic enhancer in the Suppressor of cytokine signaling 3 (*Socs3*) gene (Fig. 5f)[47], the promoter of the Nuclear paraspeckle assembly transcript 1 (*Neat1*) long non-coding RNA gene (Fig. 5g)[48], a proximal enhancer of the Cyclin D1 (*Ccnd1*) gene (Fig. 5h)[49] and the C/EBPβ promoter (Fig. 5j)[50]. Additional genes implicated in tumor progression are highlighted by these comparisons, including the Connective tissue growth factor (*Ccn2*) promoter (Fig. 5k)[51] and an intronic enhancer of the Metallothionien 2 A (*Mt2a*) gene (Fig. 5l)[52]. Finally, whereas the Testis Expressed 14 (*Tex14*) gene has not been reported to be implicated in cancer, this is the only one of the top 12 genes in which the tumor/normal differences were inconspicuous (Fig. 5i), consistent with the supposition that increases in paused RNAPII at enhancers or promoters of the other genes are associated with tumor progression.

## FFPE-CUTAC distinguishes tumor from normal tissue within the same FFPE

On-slide FFPE-CUTAC (Fig. 2a) provided us with the opportunity to compare tumor with normal tissue on the same slide. For this analysis we used ZFTA-RELA gene fusion-driven ependymomas (Fig. 6a) which are relatively large and cytologically distinct, whereas PDGFB-driven gliomas (Fig. 6b) are more diffuse. We performed on-slide FFPE-CUTAC through tagmentation and manually harvested 6 sections from a single RELA slide and 7 sections from a single PDGFB slide separately into PCR tubes. After sequencing, we performed Voom/Limma analysis comparing the sections identified cytologically as mostly tumor to sections identified as mostly normal. Results for RELA were very similar to those obtained comparing tumor to normal brains, whereas results for PDGFB showed fewer significant cCREs at FDR = 0.05 (Fig. 6c). Similar results were obtained with two other RELA slides, where the top upregulated cCRE was within the *Col1a* gene (Fig. 6d and Supplementary Data 3), which was also the top RELA-versus-Normal hit in multiple-slide comparisons (Fig. 4d). Interestingly, the top down-regulated gene in both replicate slides, *Mir124a-1hg*, is a microRNA methylation marker locus for *Helicobacter pylori* infection that correlates with gastric cancer driver gene methylation[53]. The entire locus is embedded in a cluster of 27 cCREs, and all replicates show a broad RNAPII signal in normal tissue but not RELA-driven tumor encompassing the entire cluster (Fig. 6d). Indeed, the top 10 down-regulated cCREs are either *Mir124a-1hg* or *Mir124a-2hg* and these together with the next down-regulated cCRE, which is over the *Mir670* microRNA locus, account for 15 of the top 25 down-regulated cCREs (Supplementary Data 3). In contrast, these genes are far down the RNA-seq list ranked by false discovery rate, as

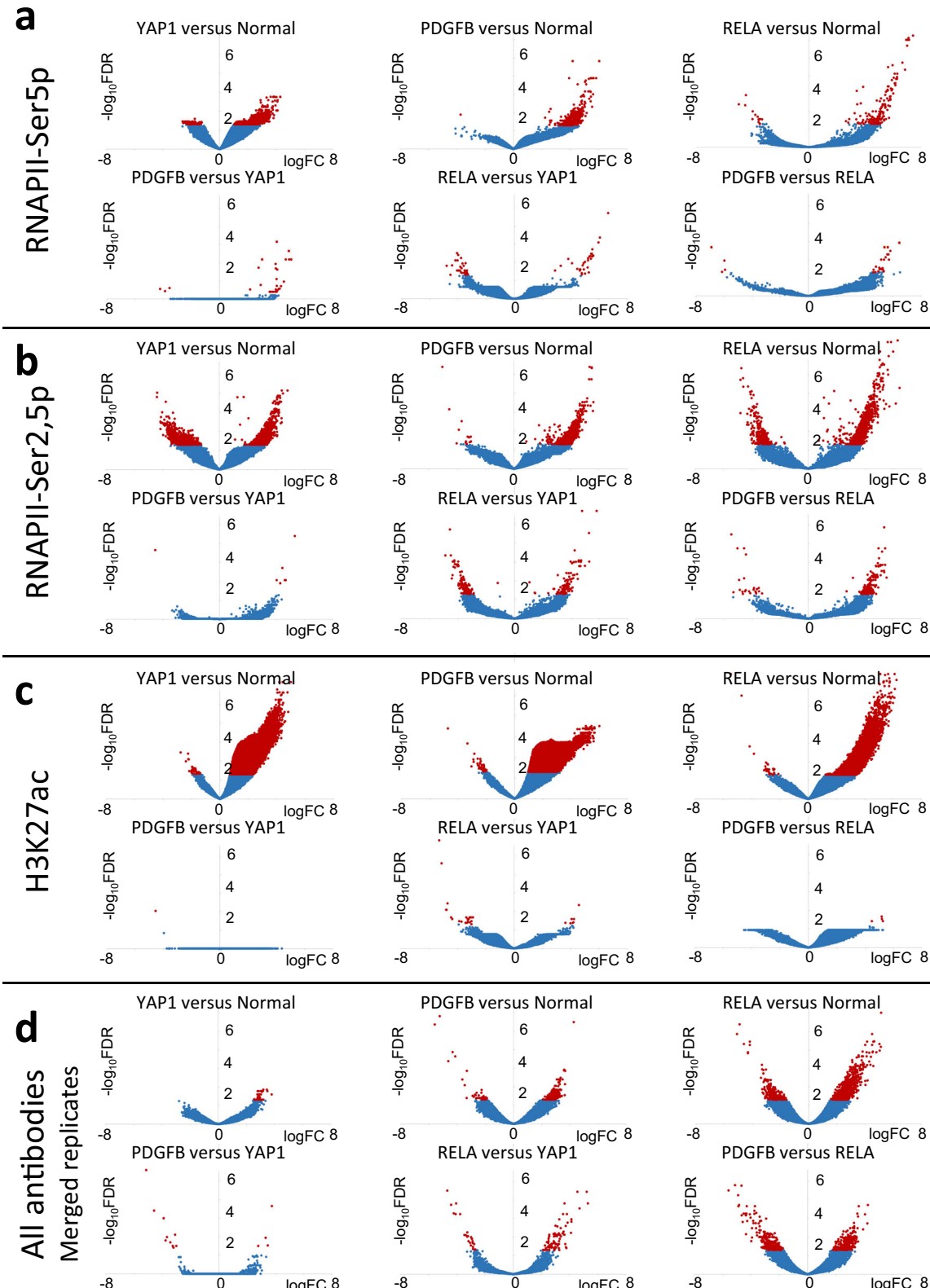

*Mir124a-1hg* ranks 9,913, *Mir124a-2hg* ranks 6,045 and *Mir670* ranks 21,262 of 23,551 annotated mouse genes (Supplementary Data 4).

## FFPE-CUTAC distinguishes tumors from normal liver

To test whether our results with mouse brain FFPEs generalize to a very different tissue type, we performed FFPE-CUTAC using FFPE sections prepared from intrahepatic cholangiocarcinoma tumors and normal liver. We used FFPE sections that had been fixed in formalin for 7 days and after deparaffinization were incubated at 90 °C in cross-link reversal buffer for 8 h and incubated with a 50:50 mixture of RNAPII-Ser5p and RNAPII-Ser2,5p antibodies, each at 1:50 concentration. Highly consistent results were obtained for samples ranging from 10% to 50% of a section (~30,000–150,000 cells), with clean peaks over housekeeping genes for both liver tumor and normal liver (Fig. 7a–d).

**Fig. 4 | Volcano plots for pairwise comparisons between FFPE-CUTAC samples.** The Degust server (https://degust.erc.monash.edu/) was used with Voom/Limma defaults to generate volcano plots, where replicates consisted of a mix of samples run in parallel or on different days on FFPE slides from 8 different brain samples (3 Normal, 3 YAP1, 1 PDGFB, 1 RELA). Input for each sample was 10–25% of an FFPE slide, which ranged from ~50,000-100,000 cells per 10-micron section. **a** Comparisons based on RNAPII-Ser5p using average normalized counts per base-pair for each cCRE, applying the Empirical Bayes Voom/Limma algorithm for pairwise comparisons using the other datasets as pseudo-replicates to increase statistical power. Replicate numbers: Normal: 13; YAP1: 14, PDGFB: 3; RELA: 2.

**b** Same as (**a**) for RNAPII-Ser2,5p. Replicate numbers: Normal: 5; YAP1: 6; PDGFB: 3; RELA: 3. **c** Same as (**a**) for H3K27ac. Replicate numbers: Normal: 10; YAP1: 12; PDGFB: 5; RELA: 7. **d** Datasets from multiple FFPE-CUTAC experiments for each antibody (RNAPII-Ser5p, RNAPII-Ser2,5p or H3K27ac) or antibody combination (RNAPII-Ser5p + RNAPII-Ser2,5p) were merged, then down-sampled to the same number of mapped fragments for each genotype. These 16 datasets (4 antibodies x 4 genotypes) were compared against each other with Voom/Limma using the other 14 datasets as pseudo-replicates. Top hits FDR < 0.05 (red) are listed in Supplementary Data 5.

As was the case with brain tumor and normal tissues fixed in formalin for 2 days, the number of peaks and fraction of reads in peaks (FRiP) were much higher than those from FACT-seq FFPE livers (Fig. 7e, f) and overlap with cCREs was also much higher when down-sampled to the same number of fragments (Fig. 7g). Finally, volcano plots revealed net increases in cCRE RNAPII occupancy both in fold-change and FDR for liver tumors relative to normal livers, similar to what we observed in comparing brain tumors to normal brains (Fig. 7h, i). We conclude that FFPE-CUTAC provides high-quality data for FFPEs from diverse tissue types.

## Comparison between FFPE-CUTAC and standard RNA-seq on transgene-driven brain tumors

The murine brain tumors that we used in our study have served as models for the study of de novo tumorigenesis[28,29,54], with high-quality RNA-seq data available. To do an unbiased comparison between FFPE-CUTAC regulatory elements and processed transcripts mapped by RNA-seq, we first determined whether there is sufficient overlap between cCREs and annotated 5′-to-3′ genes to fairly compare these very different modalities. Specifically, the 343,731 cCREs average 272 bp in length, accounting for 3.4% of the Mm10 build of the mouse genome, whereas the 23,551 genes in RefGene average 49,602 bp in length, with an overlap of 54,062,401 bp or 2.0% of Mm10. In other words, the 5′-to-3′ span of mouse genes on the RefGene list should capture all of the RNA-seq true positives and almost 60% (2.0/3.4 x 100%) of the cCREs. With most cCREs overlapping annotated mouse genes, we can directly compare FFPE-CUTAC fragment counts to RNA-seq fragment counts by asking how well they correlate with one another over genes. Whereas FFPE-CUTAC replicates and RNA-seq replicates are very strongly correlated to a similar extent, with "arrowhead" scatterplots ($R^2 = 0.955–0.997$), comparisons between FFPE-CUTAC and RNA-seq samples are "fuzzy" but nevertheless show strong correlations ($R^2 = 0.764–0.881$) (Fig. 8a).

We also determined the extent to which the same genes differ significantly between tumor and normal in the two datasets. Using an FDR = 0.05 cut-off for both FFPE-CUTAC and RNA-seq, we found that 80–82% of genes were found in both lists: 52 of 63 for YAP1-driven tumors versus normal brains, 268 of 336 for PDGFB-driven versus normal and 1519 of 1896 for RELA-driven versus normal. However, there is a striking difference in the specificity with which these genes are identified as illustrated by comparison of volcano plot displays: FFPE-CUTAC provides high specificity for regulatory elements, where significant differences between cCREs are almost exclusively at the upregulated corner of the volcano plots (high positive $\log_2$ fold-change, high $-\log_{10}$ FDR) (Fig. 4). In contrast, about 1/3 to 1/2 of 23,551 genes show significant differences between these tumors and normal brains using RNA-seq with massive, mostly symmetrical "volcanic eruptions" (Supplementary Fig. 4).

To validate these comparisons, we aligned profiles of FFPE-CUTAC and RNA-seq at YAP1 and at nine direct targets of YAP1, which were previously determined based in part on the RNA-seq data[54]. As expected, the FFPE-CUTAC profiles are enriched primarily at 5′ ends and RNA-seq at 3′ ends (Fig. 8b). Importantly, all ten examples showed full or partial concordance between FFPE-CUTAC and RNA-seq. We

conclude that there is overall excellent agreement between our FFPE-CUTAC data and previously published high-quality RNA-seq datasets. The very high specificity of FFPE-CUTAC data and its ability to identify and map differentially regulated microRNAs, together with its simple implementations and potential for automation, make it an exceptional modality for discovery of functional biomarkers.

## Discussion

Fixation-related DNA and chromatin damage has thus far impeded the practical application of chromatin profiling to FFPEs[4]. Here we have shown that improvements to the single-tube CUT&Tag-direct protocol to make it suitable for whole cells, and together with heat treatment of FFPEs, provides high-quality CUTAC data. Using RNAPII antibodies provides a ground-truth interpretation of active chromatin based on the transcriptional machinery itself, applicable to both promoters and enhancers[55]. RNAPII-based CUTAC mapping contrasts with the mapping of "open chromatin" inferred from enzymatic [e.g. DNAseI hypersensitivity mapping[56] and ATAC-seq[11]] or physical [e.g. FAIRE[15] and Sono-seq[57]] methods, where results typically differ depending on the method used. We also showed that all FFPE-CUTAC steps through tagmentation can be performed on the slides without using organic chemicals such as xylene or mineral oil. On-slide FFPE-CUTAC allows for direct comparisons between dissected tumor and normal tissues from the same FFPE 5- or 10-micron section. As all steps through tagmentation are performed on the slide without noticeable tissue disruption (Fig. 6a, b), FFPE-CUTAC is suitable for spatial applications using available platforms[58,59].

RNA-seq has been the preferred method for profiling the transcriptome, however, it is strongly biased towards abundant transcripts, while transcription factors that drive development and are deregulated in cancer may be expressed at relatively low levels and can be difficult to detect. Changes in TF mRNA abundance with cell type changes are swamped out by changes in more abundant mRNAs. mRNA abundance is also affected by complex regulatory mechanisms occurring during and after transcriptional elongation, including multiple modes of processing and export to the cytoplasm, resulting in undifferentiated volcano plots that resemble erupting volcanos (Supplementary Fig. 4). Our comparisons were against RNA-seq data from fresh or frozen tissue, whereas RNA-seq results from FFPEs are much poorer owing to serious degradation and off-target reads[60]. In contrast, paused RNAPII is a critical checkpoint for transcriptional activation, and so its abundance at a regulatory element is a direct measure of transcriptional competence, and cancer driver and tumor suppressor loci stand out (Figs. 5 and 6c, d). The 343,731 genomic sites annotated as candidate cis-regulatory elements (cCREs) in the mouse genome can potentially provide direct information on transcriptional regulatory networks. We found that FFPE-CUTAC sensitively detects cCRE clusters spanning microRNA loci that gain RNAPII in the RELA fusion-driven tumor. Strikingly, the 10 most down-regulated cCREs corresponded to two unlinked loci for the mouse Mir124a microRNA, which was previously described as a neuronal differentiation factor[61], a tumor suppressor in brain[62] and a unique human biomarker for H. pylori infection and gastric cancer risk[53]. Although Mir124a is one of the most abundantly expressed microRNAs in the central nervous

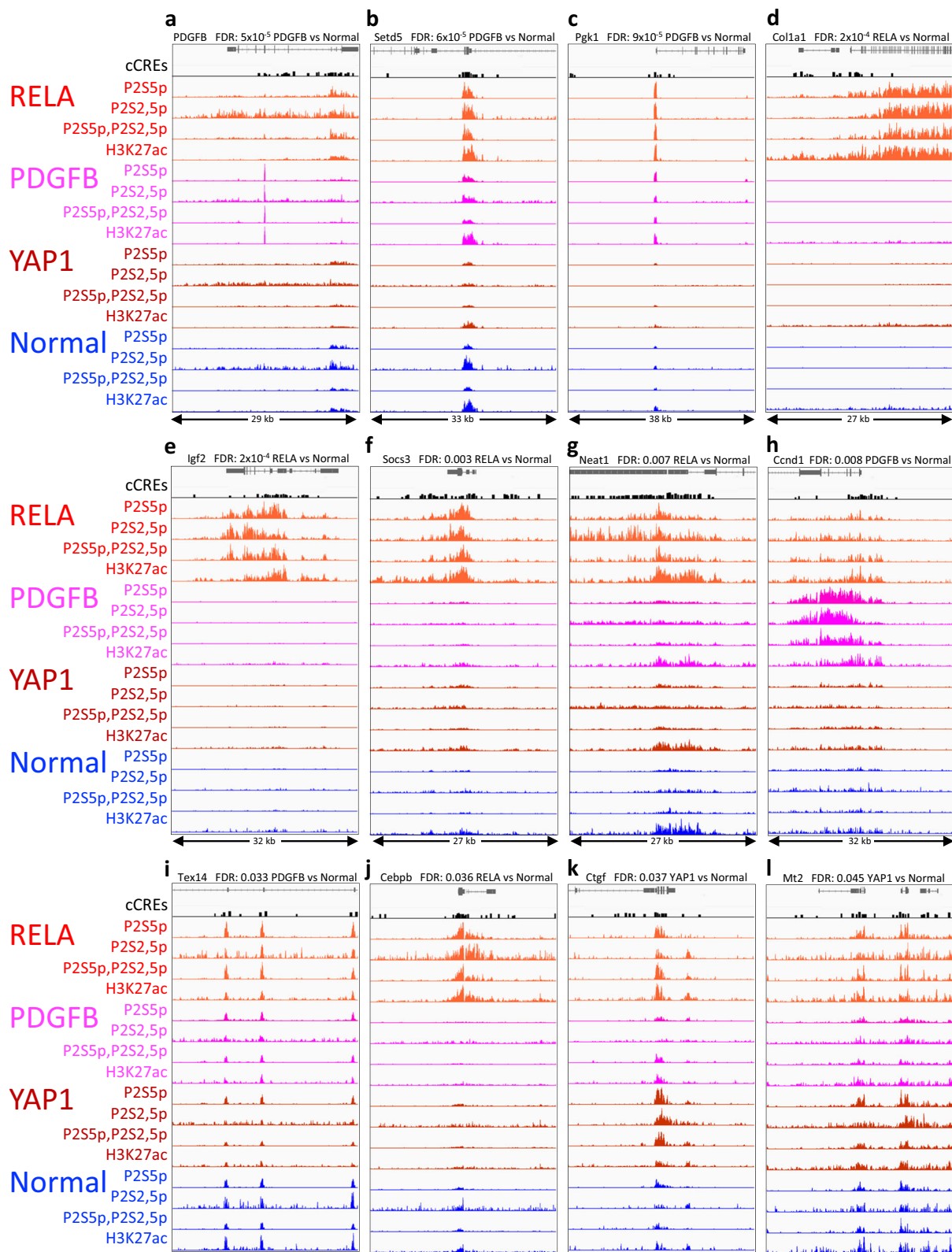

**Fig. 5 | Top significant differences between tumor and normal and between tumors based on RNAPII-Ser5p FFPE-CUTAC comparisons. a–e** IGV tracks centered around the cCREs with the most significant difference across all pairwise comparisons (FDR = $5 \times 10^{-5}$ – $2 \times 10^{-4}$). To enrich for regulatory elements within the span of each cCRE, we used the maximum value. **f–l** Tracks centered around the cCRE for each of the strongest signals with FDRs < 0.05, ordered by increasing FDR (0.003–0.045).

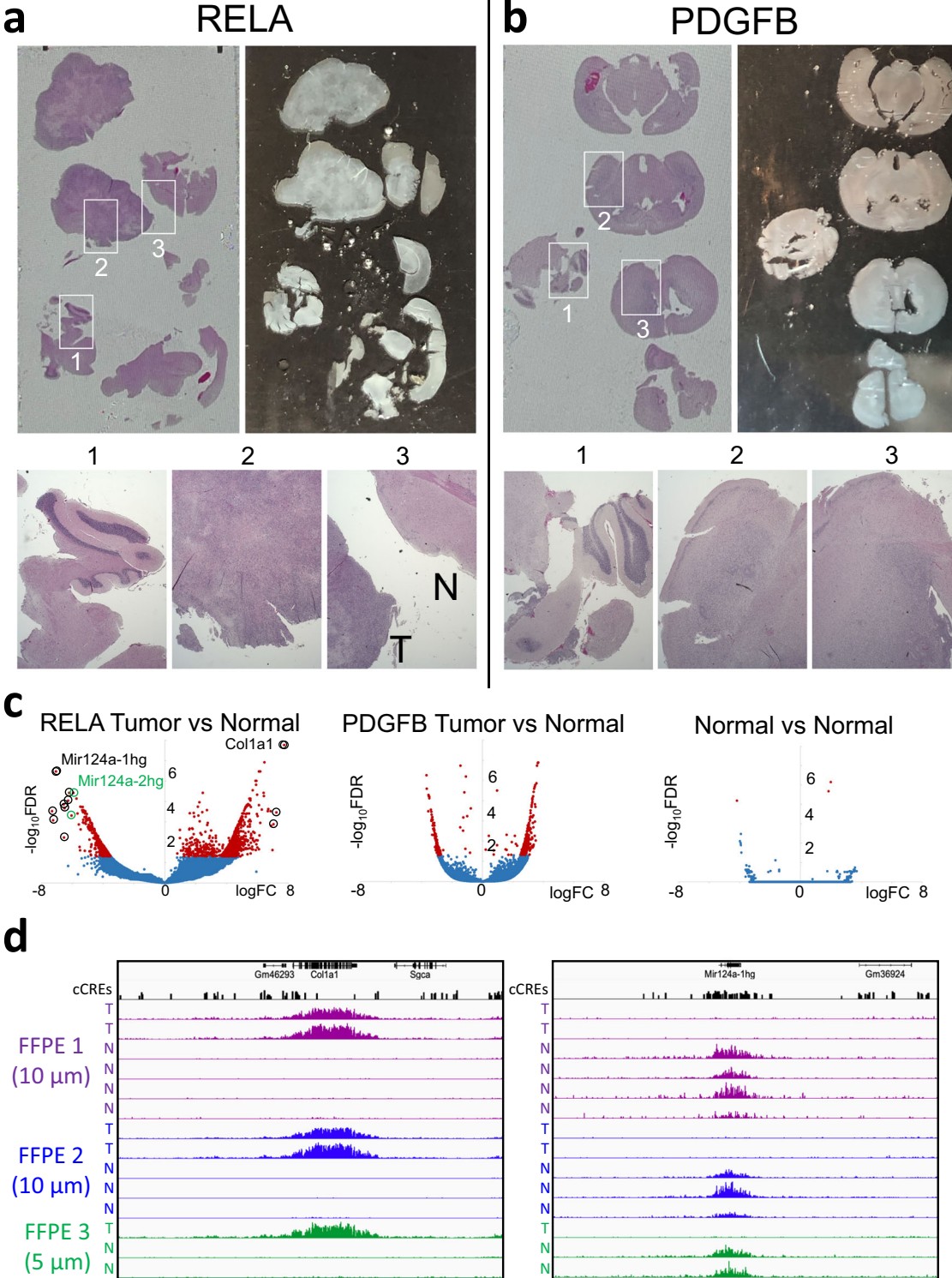

**Fig. 6 | FFPE-CUTAC distinguishes tumor from normal tissue within the same FFPE section. a** RELA drives well-defined ependymomas where dissection following tagmentation and transfer of whole sections to PCR tubes after RNAPII-Ser5p FFPE-CUTAC post-tagmentation successfully separated tumor from normal tissue with volcano plot results similar to that for RELA versus Normal brains (Fig. 4). **b** In contrast, PDGFB-driven gliomas are relatively diffuse, and separation of sections post-tagmentation resulted in fewer significant target cCREs. **c** Left: Volcano plot for FFPE Slide 1 (2 tumor 4 normal sections) using two other slides with 3 tumor and 5 normal sections as pseudo-replicates. Top RELA FFPE hits based on FDR < 0.01 and greatest fold-change are circled and tabulated in (Supplementary Data 6). Middle: Volcano plot for a PDGFB slide (3 tumor, 4 normal). Right: Volcano plot for a normal brain (5 versus 5 replicates). Replicate tracks for the two top upregulated (*Col1A1*) and down-regulated (*Mir124a-1hg*) loci are shown, group-autoscaled, where red marks dots with FDR < 0.05. **d** Tracks for the RELA Tumor-versus-Normal experiments are shown for *Col1a1* (left) and *Mir124a-1hg* (right) color-coded and group-autoscaled for each replicate FFPE slide dissected.

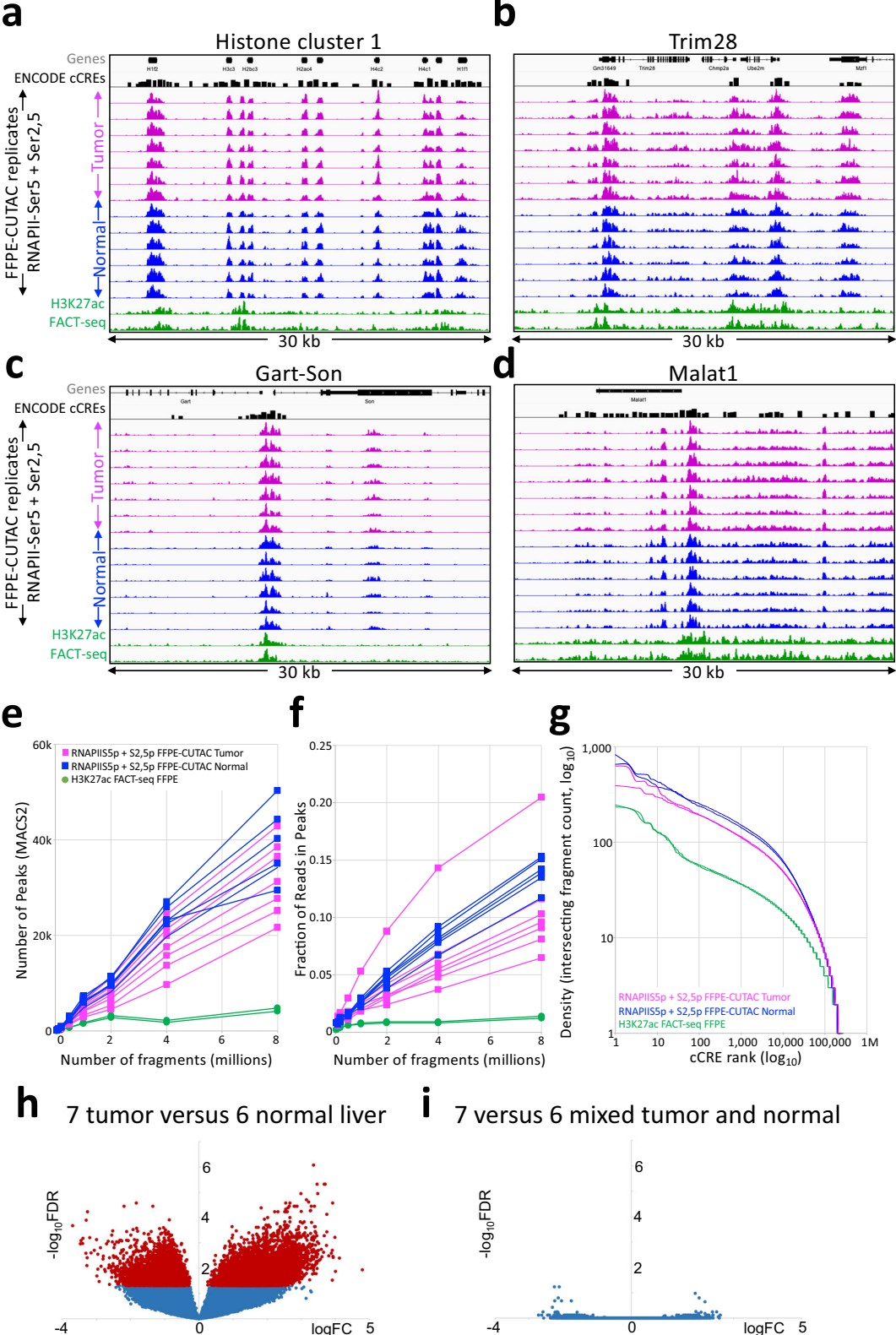

**Fig. 7 | FFPE-CUTAC produces high-quality data from liver FFPEs.**
**a**–**d** Representative tracks of liver tumor and normal liver FFPE-CUTAC and FACT-seq samples at the housekeeping gene regions depicted in Fig. 3. A track for Candidate *cis*-Regulatory Elements (cCREs) from the ENCODE project is shown above the data tracks, which are autoscaled for clarity. **e**, **f** Number of peaks and Fraction of Reads in Peaks (FRiP) called using MACS2 on samples containing the indicated number of cells for 7 liver tumor (magenta), 6 normal liver (blue) and 2 normal liver FACT-seq (green) samples. **g** Cumulative $\log_{10}$ plots of normalized counts intersecting cCREs versus $\log_{10}$ rank for representative liver samples, where red marks dots with FDR < 0.05. **h** Voom/Limma volcano plot for the 7 liver tumors versus 6 normal liver samples. **i** Control volcano plot in which three liver tumor samples and 3 normal livers were exchanged for Voom/Limma analysis. Rank-ordered cCREs based on averages are tabulated in Supplementary Data 7. Source data are provided as a Source Data file.

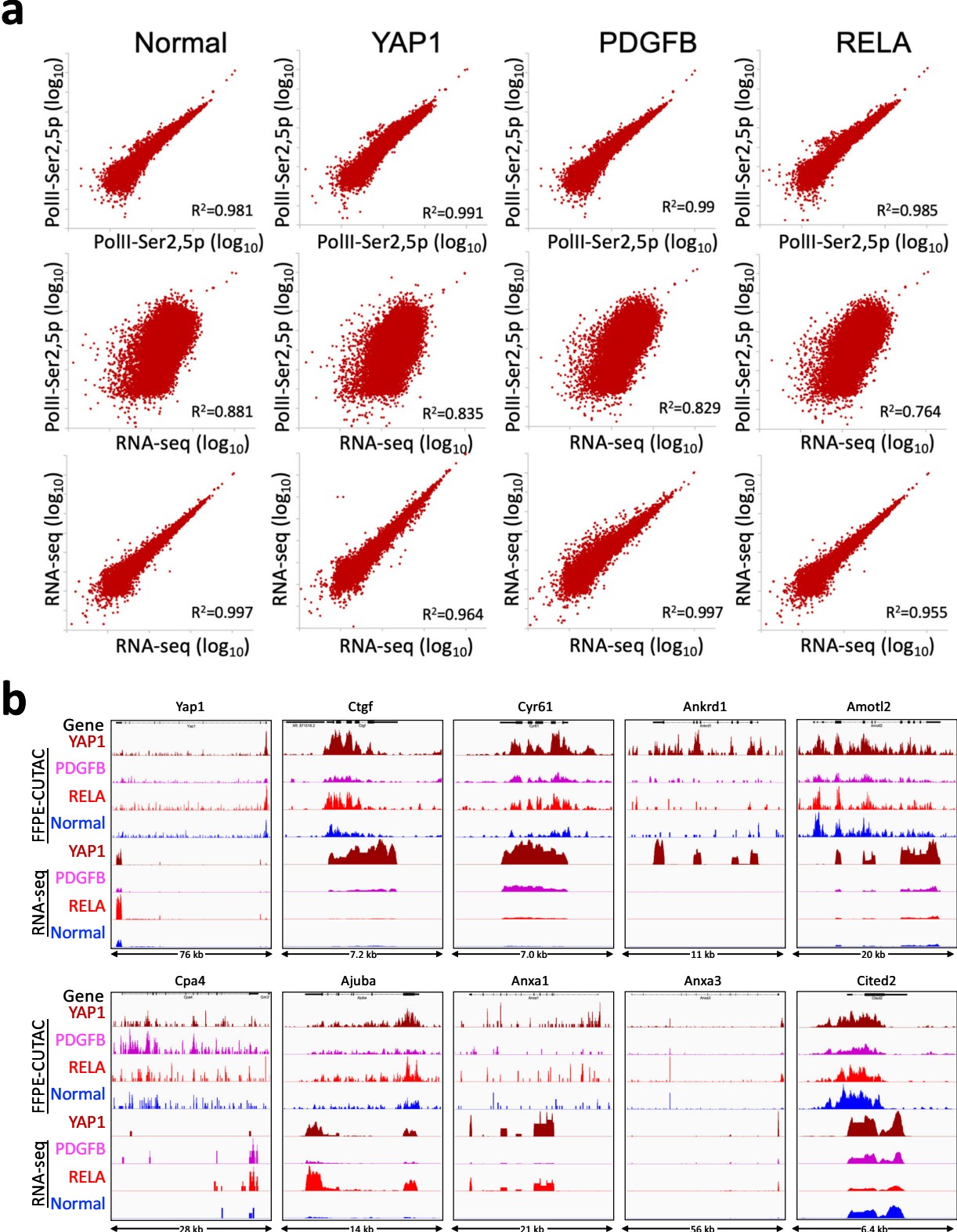

**Fig. 8 | Comparisons between FFPE-CUTAC and RNA-seq. a** Top panels: Scatterplots of representative FFPE-CUTAC replicate samples from RNAPII-Ser2,5p for normal brain and the three tumors. Middle panels: Scatterplots of comparisons between the RNAPII-Ser2,5p sample and the corresponding RNA-seq dataset. Lower panel: Scatterplots of RNA-seq datasets used in the comparisons. **b** Comparisons between FFPE-CUTAC and RNA-seq over Yap1 and previously reported Yap1 direct targets. Tracks were group-autoscaled within modalities. Rank-ordered pairwise comparisons are listed in Supplementary Data 4.

system[61], we found that it is present at only average levels in RNA-seq data. As microRNAs are excised from polyadenylated RNAPII transcripts, and RNA-seq relies on priming from the poly(A) tail, important microRNA biomarkers such as Mir124a are entirely missed. In contrast to RNA-seq, FFPE-CUTAC detects all classes of active RNAPII genes. Furthermore, the much better discrimination of RNAPII that we observed for FFPE-CUTAC over cCREs than for high-quality RNA-seq data over genes encourages more general application of FFPE-CUTAC technology for diagnosis, biomarker discovery and retrospective studies.

Remarkably, the large majority of significant differences between tumors and normal brain corresponded to increases in RNAPII and H3K27ac, a histone mark of active promoters and enhancers. Global hypertranscription is a general feature of aggressive human cancers[63], and our consistent finding of greater upregulation of RNAPII at cCREs in tumors relative to normal, even from the same mouse brains, provides support for this interpretation. In brain tumors, the RNAPII FFPE-CUTAC fragment size distribution was increased relative to that of normal tissue, perhaps indicative of greater accessibility to the transcriptional apparatus.

Cross-links and adducts resulting from the long incubations in formaldehyde necessary for long-term preservation cause DNA breaks and lesions that are serious impediments for most genomic methods applied to FFPEs. Indeed, standard CUT&Tag failed for the group that developed FACT-seq[20], and we also failed to obtain usable profiles for repressive H3K27me3 and H3K9me3 and gene-body H3K36me3 histone epitopes. We attribute these failures to the tight wrapping of DNA around lysine-rich histones, which are the most susceptible to cross-linking and formation of DNA adducts that result in DNA breaks during high-temperature cross-linking reversal[24]. In contrast, nucleosome-depleted regions (NDRs) that are mapped using accessibility methods such as ATAC-seq[11], NicE-seq[13], FAIRE[14,15] and CUTAC[22,23] are much better suited for FFPEs, as the protein machineries that occupy these sites are not especially lysine-rich. In particular, the YSPTSPS heptamer present in 52 tandem copies on the C-terminal domain of the largest subunit of RNAPII presents abundant lysine-free epitopes for CUT&Tag, and the use of low-salt tagmentation after stringent washes allows for tight binding of the Tn5 transposome within the confines of the NDR. We have previously shown that for epitopes such as H3K4 methylations[22] and RNAPII epitopes[23] that flank gaps in the nucleosome landscape at promoters and enhancers, tagmentation preferentially releases subnucleosomal fragments. FACT-seq improves yield with in vitro transcription from a T7 promoter inserted at single sites, however this strategy foregoes the advantage of the small size of NDRs at promoters and enhancers where nevertheless two Tn5s can fit with enough DNA in between for sequence-based mapping. We might attribute the better data quality that we obtained using CUTAC relative to FACT-seq to the very low probability of two Tn5s inserting close enough to one another and correctly oriented to produce a small amplifiable fragment by random chance. Curiously, H3K27ac FFPE-CUTAC detected cCREs even more sensitively than standard H3K27ac CUT&Tag on frozen tissue, which might indicate that better reversal of cross-links at NDRs than at nucleosomes facilitates tagmentation within NDRs while nucleosomes remain relatively intractable. Indeed, by avoiding the use of degradative enzymes and using only heat to expose epitopes in a suitable buffer, we found that bead-bound tissue shards from sheared FFPEs are much easier to handle without damage than cells or nuclei, where lysis and sticking is a constant concern.

We also discovered that DNA from *Rhodococcus erythropolis*, a species of bacteria that can live on paraffin wax as its only carbon source, is abundant in the FFPE samples that we processed, and this unfixed DNA competes against formalin-damaged DNA from FFPEs during PCR. We found that bacterial contamination was essentially eliminated in on-slide FFPE-CUTAC and when using some magnetic beads that lacked ConA but nevertheless bound sufficiently well to

FFPE tissue fragments. These observations suggest that sugars on the surface of contaminating bacterial remnants were captured by the Concanavalin A sugar-binding site and their DNA was released prior to or during PCR. As CUT&Tag-direct has been fully automated[26], we expect that our FFPE-CUTAC protocol will be suitable for institutional core facilities and commercial services, maximizing reproducibility and minimizing costs.

In conclusion, we have shown that RNAPII and H3K27ac chromatin profiling can be conveniently and inexpensively performed on FFPEs in single PCR tubes or directly on slides. We use only heat in a suitable buffer to reverse the cross-links while making the tissue sufficiently permeable, followed by modified versions of our CUT&Tag-direct protocol, which is routinely performed in many laboratories[23,64]. We found that data quality using low-salt tagmentation for antibody-tethered chromatin accessibility mapping is sufficient to distinguish cancer from normal tissues and resolve closely similar brain tumors. Using FFPE-CUTAC, our study identified direct targets of cancer drivers in tumors and microRNA loci not detectable by RNA-seq, validating our approach.

## Methods

### Ethical statement
This research was approved by the Fred Hutch Institutional Animal Care and Use Committee (Protocol # 50842) and complies with all required ethical regulations.

### Cell lines
Human female K562 chronic myelogenous leukemia cells (American Type Culture Collection (ATCC) cat. no. CCL-243) and mouse NIH 3T3 cells (ATCC cat. no. CRL-1658) were authenticated for STR, sterility, human pathogenic virus testing, mycoplasma contamination and viability at thaw. H1 (WA01) male hESCs (WiCell cat. no. WA01-WB35186) were authenticated for karyotype, STR, sterility, mycoplasma contamination and viability at thaw. K562 cells were cultured in liquid suspension in IMDM (ATCC) with 10% FBS added (Seradigm). H1 cells were cultured in Matrigel (Corning)-coated plates at 37 °C and 5% $CO_2$ using mTeSR-1 Basal Medium (STEMCELL Technologies) exchanged every 24 h. K562 and 3T3 cells were harvested by centrifugation for 3 minutes at 1,000 g and then resuspended in 1× PBS. H1 cells were harvested with ReleasR (STEMCELL Technologies) using the manufacturer's protocols.

### Mice
All animal experiments were done in accordance with protocols approved by the Institutional Animal Care and Use Committees of Fred Hutchinson Cancer Center (protocol no. 50842) and followed National Institutes of Health guidelines for animal welfare. The RCAS/tv-a system used in this work has been described previously[54]. In brief, Jackson lab mouse strain 3529 (FVB/N;C57BL/6;129/Sv Nestin) (N)/tv-a Cdkn2a null pups (P0-P1; male and female) or adults (5–7 week old, male and female) had been injected intracranially with DF-1 cells expressing RCAS-PDGFB[27], RCAS-REL A-ZFTA[28], or RCAS-YAP1-FAM118b[65]. We used both male and female mice. Upon weaning (-P21), mice were housed with same-sex littermates, with no more than 5 per cage and given access to food/water ad libitum and monitored daily for the occurrence of tumor-related symptoms for the duration of the experiment. Mice were euthanized upon the occurrence of predefined tumor-related symptoms: macrocephaly, lethargy, dehydration, poor grooming, hemiparesis, weight loss, seizures, jumpiness, or immobilization/paralysis.

### Mouse tumor and normal tissues and FFPEs
Ntva;cdkn2a-/- mice were injected intracranially with DF1 cells infected with and producing RCAS vectors encoding either PDGFB[27], REL A-ZFTA[28], or YAP1-FAM118b[65]. When the mice became lethargic and

showed poor grooming, they were euthanized and their brains removed and fixed at least 48 h in Neutral Buffered Formalin. Tumorous and normal brains were sliced into five pieces and processed overnight in a tissue processor, mounted in a paraffin block and 5- or 10-micron sections were placed on slides. Slides were stored for varying times between 1 month to ~2 years before being deparaffinized and processed for FFPE-CUTAC. Healthy mouse liver or intrahepatic cholangiocarcinomas tumors harvested from orthotopic models of intrahepatic cholangiocarcinoma mice with activating mutations of Kras[G12D] and deletion of p53[66] were fixed in formalin for 7d before being sent to the Fred Hutch Experimental Histopathology Shared Resource for FFPE processing. Ten brains and five livers were used in the experiments described.

### Antibodies
Primary antibodies: H3K4me3: Active Motif cat. no. 39159, lot no. 18122006; H3K27ac: Abcam cat. no. ab4729, lot no. 1033973; RNAPII-Ser5p: Cell Signaling Technologies cat. no. 13523, lot 3; RNAPII-Ser2,5p: Cell Signaling Technologies cat. no. 13546, lot 1; H3K27me3: Cell Signaling Technologies cat. no. 9733, lot 19; H3K4me2: Epicypher cat. no. 13-0027, lot 21090003-01; H3K36me3: Thermo cat. no. MAS-24687, lot VE2997961. Secondary antibody: Guinea pig α-rabbit antibody (Antibodies online cat. no. ABIN101961, lot 46671).

### CUT&Tag-direct for whole cells
Concanavalin A (ConA) coated magnetic beads (Bangs Laboratories, cat. no. BP531) were activated just before use with Ca++ and Mn++ as described[21]. Frozen whole-cell aliquots were thawed at room temperature, split into PCR tubes and 5 μL ConA beads were added with gentle vortexing. Briefly, nuclei were mixed with activated Concanavalin A beads and resuspended in Triton-wash buffer (20 mM HEPES pH 7.5, 150 mM NaCl, 0.5 mM spermidine, 0.05% Triton-X100 and Roche EDTA-free protease inhibitor). After successive incubations with primary antibody (≥1 h) and secondary antibody (1 h) in Wash buffer, the beads were washed and resuspended in pAG-Tn5 preloaded with mosaic end adapters (Epicypher cat. no. 15-1117 1:20) in Triton-wash buffer for 1 h. Incubations were done at room temperature in 25 μL volumes in PCR tubes. Tagmentation was performed for 1 h in 10 mM TAPS pH 8.5, 20% N,N-dimethylformamide, 5 mM MgCl₂ at 55 °C. Fragment release was performed in 5 μl 1% SDS supplemented with 1:10 Thermolabile Proteinase K (New England Biolabs cat. no. P8111S) at 37 °C 1 h followed by 58 °C 1 h. SDS was quenched by addition of 15 μl 6% Triton-X100 and PCR was performed by addition of 2 μl each barcoded 10 mM i5 and i7 primer solutions and 25 μl NEBNext 2X PCR Master mix (New England Biolabs cat. no. ME541L) (Supplementary Data 8) The following cycling conditions were used: Cycle 1: 58 °C for 5 min; Cycle 2: 72 °C for 5 min; Cycle 3: 98 °C for 5 min; Cycle 4: 98 °C for 10 s; Cycle 5: 60 °C for 10 s; Repeat Cycles 4-5 11 times; 72 °C for 1 min; Hold at 8 °C. Clean-up was performed using HighPrep PCR Cleanup Magbio Genomics cat. no. AC-60500 following manufacturer's instructions. A detailed step-by-step protocol is available at Protocols.io: https://doi.org/10.17504/protocols.io.x54v9mkmzg3e/v4.

### FFPEs
Mouse tissue (including normal brains and tumor bearing brains) were removed, fixed in 10% neutral-buffered formalin for a minimum of 24 h and embedded into paraffin blocks. 5- or 10-μm serial sections were cut from formalin-fixed paraffin-embedded specimens and mounted on slides.

### FFPE-CUTAC
In most experiments, deparaffinization was performed in Coplin jars using 2-3 changes of histology grade xylene over a 20-minute period, followed by 3-5 minute rinses in a 50:50 mixture of xylene:100%

ethanol, 100% ethanol (twice), 95% ethanol, 70% ethanol and 50% ethanol, then rinsed in deionized water. Slides were stored in distilled deionized water containing 0.02% sodium azide for up to 2 weeks before use. In experiments presented in Fig. 6 and Supplementary Data 1 and 6, FFPE slide were placed in 800 mM Tris-HCl pH8.0 in a slide holder and incubated at 85 °C for 8-16 h, whereupon the paraffin melted and floated off the slide. Liquid was added beneath the surface so that any residual paraffin would drain out over the top of the slide holder.

Tissue sections on deparaffinized slides were diced using a razor and scraped into a 1.7 mL low-bind tube containing 400 μl 800 mM Tris-HCl pH8.0, 0.05% Triton-X100. For xylene-deparaffinized samples, incubations were performed at 80–90 °C for 8–16 h or as otherwise indicated either in a heating block or divided into 0.5 mL PCR tubes after needle extraction. Needle extraction was performed either before or after Concanavalin A (ConA) bead addition using a 1 ml syringe fitted with a 1" 22 gauge needle with 20 up-and-down cycles, and in some cases was followed by 10 cycles with a 3/8" 26 gauge needle. In some experiments amine-coated (Polysciences cat. no. 86001-10) or glutathione-coated (Fisher cat. no. 88822) paramagnetic beads were used in place of ConA beads. Other steps through to library preparation and purification followed the CUT&Tag-direct protocol as described above. A detailed step-by-step protocol, is available on Protocols.io: https://doi.org/10.17504/protocols.io.14egn292zg5d/v1, with a comment box for help.

### On-slide FFPE-CUTAC
FFPE slide were placed in 800 mM Tris-HCl pH8.0 in a slide holder and incubated at 85 °C for 8-16 h, whereupon the paraffin melted and floated off the slide. Slides were cooled to room temperature, dipped in Triton-Wash buffer (10 mM HEPES pH 7.5, 150 mM NaCl, 2 mM spermidine and Roche complete EDTA-free protease inhibitor), drained and excess liquid wicked off using a Kimwipe tissue. The sections were immediately covered with 50-100 μL primary antibody added dropwise. Plastic film was laid on top starting at the bottom end and omitting bubbles as the meniscus progressed toward the frosted end of the slide. The excess plastic film was folded under for a near-watertight seal. After ≥2 h incubation at room temperature (or overnight at ~8 °C) in a moist chamber, the plastic film was peeled back, and the slide was submerged in Triton-Wash buffer for 10-20 min. This incubation/wash cycle was repeated for the guinea pig anti-rabbit secondary antibody (Antibodies Online cat. no. ABIN101961) and for pAG-Tn5 preloaded with mosaic end adapters (Epicypher cat. no. 15-1117 1:20), followed by transfer of the slide to 10 mM TAPS pH 8.5. Tagmentation was performed in 5 mM MgCl₂, 10 mM TAPS pH 8.5, 20% (v/v) N,N-dimethylformamide in slide holders incubated at 55 °C for 1 h. Following tagmentation, slides were dipped in 10 mM TAPS pH 8.5, drained and excess liquid wicked off. Individual sections were covered with 2 μL 10% Thermolabile Proteinase K (TL ProtK) in 1% SDS using a pipette tip to loosen the tissue. Tissue was transferred to a thin-wall PCR tube containing 2 μL TL ProtK using a watchmaker's forceps, followed by 1 μL TL ProtK and transfer to the PCR tube. Tubes were incubated at 37 °C for 1 h and 58 °C for 1 h before PCR as described above. A detailed step-by-step protocol, is available on Protocols.io https://doi.org/10.17504/protocols.io.14egn292zg5d/v1.

### DNA sequencing and data processing
The size distributions and molar concentration of libraries were determined using an Agilent 4200 TapeStation. Up to 48 barcoded 96 barcoded libraries were pooled at approximately equimolar concentration for sequencing. Paired-end 50x50 bp sequencing on the Illumina NextSeq 2000 platform was performed by the Fred Hutchinson Cancer Center Genomics Shared Resources. This yielded 1–20 million reads per antibody. Adapters were clipped by cutadapt version 4.1 with parameters --nextseq-trim 20 -m 20 -a

AGATCGGAAGAGCACACGTCTGAACTCCAGTCA -A AGATCGGAA-
GAGCGTCGTGTAGGGAAAGAGTGT -Z

Clipped reads were aligned by Bowtie2 version 2.4.4 to the *Mus musculus* mm10 and *Homo sapiens* hg19 reference sequences from UCSC and to the *Rhodococcus erythropolis* complete genome (NZ_CP007255.1) from NCBI with parameters --very-sensitive-local --soft-clipped-unmapped-tlen --dovetail --no-mixed --no-discordant -q --phred33 -I 10 -X 1000

## Data analysis

BLASTN searches of unmapped reads against the Nucleotide database were done on the NCBI web site (https://blast.ncbi.nlm.nih.gov/Blast.cgi?PROGRAM=blastn&PAGE_TYPE=BlastSearch&LINK_LOC=blasthome). We noticed the majority hit several bacteria so we narrowed the search to the RefSeq Genome Database restricted to Bacteria (taxid:2). After further analysis of these BLAST hits we made a Bowtie2 reference sequence from five bacteria: NZ_CP007255.1 *Rhodococcus erythropolis* R138

NZ_JACNZU010000010.1 *Bacillus pumilus* strain 167T-6
NZ_JAGEKP010000001.1 *Leifsonia* sp. TF02-11
NZ_QCYC01000100.1 *Vibrio vulnificus* strain Vv003
NZ_MLHV01000015.1 *Mycobacterium syngnathidarum* strain 24999

To estimate library sizes in Supplementary Data 1, we used Picard Tools: MarkDuplicates http://broadinstitute.github.io/picard/

Properly paired reads were extracted from the alignments by samtools version 1.14 bamtobed command into mapped fragment bed files and normalized count tracks were made by bedtools version 2.30[67,68] genomecov command with scale (size_of_reference_sequence/total_counts). Normalized count tracks are the fraction of counts at each base pair scaled by the size of the reference sequence so that if the counts were uniformly distributed across the genome there would be one at each position. Distributions of the lengths of the mapped fragments were made using the UNIX sort and uniq -c command. Peaks were made by MACS2 version 2.2.6[41] from the mapped fragment bed files with parameters:

macs2 callpeak -t <fragments > -f BEDPE -g hs --keep-dup all -p 1e-5 -n <name > --SPMR

For comparisons, the following datasets were downloaded from GEO: GSM5530653, GSM5530654 and GSM5530655 (mouse kidney H3K27ac FACT-seq replicates 1–2 and H3K27ac Frozen CUT&Tag, respectively), GSM5530669 and GSM5530670 (mouse liver H3K27ac FACT-seq replicates 1–2) and GSE172688 (ENCODE ChIP-seq mouse post-natal forebrain).

Random sub-samples of fixed sizes were taken from the mapped fragment bed files using the UNIX shuff command and peaks were found by MACS2 for each sub-sample. Then the fraction of reads in peaks (FRiP) was computed using the bedtools intersect command. Single-end data used for comparisons was 50 bp for kidney and 75 bp for liver. These read lengths were sufficiently similar to our paired-end median adapter-trimmed fragment lengths (65-76 bp for brain and 63-68 bp for liver) that no adjustments were made in comparisons.

cCRE overlaps were calculated for 10 million mapped fragments per sample as the number of fragments with at least one base pair overlap with a cCRE. Differential analyses of FFPE-CUTAC and RNA-seq data were performed using the Voom/Limma option[42] on the Degust server (https://degust.erc.monash.edu/).

Files for degust (https://degust.erc.monash.edu/) were made for a list of 343,731 Candidate *cis*-regulatory elements (cCREs) for *Mus musculus* from ENCODE (ENCFF427VRW) and for 23,551 genes from the *Mus musculus* Mm10 refGene list from the University of California Santa Cruz Genome Resource. The refGene file contains multiple transcripts for each gene so we winnowed it by using the region from the minimum start position to the maximum end position for each set

of transcripts for a gene. For sums, we added the normalized counts within each cCRE or gene region for analysis by the degust web site. For summits we took the maximum within each region.

*Statistics and Reproducibility* Overall data quality was evaluated by peak-calling and FRiP at multiple levels of downsampling and by Voom/Limma (-log₁₀FDR versus log₂FoldChange) analysis, which is very sensitive to reproducibility of replicates. No statistical method was used to predetermine sample size nor were data excluded from the analyses. The experiments were not randomized and Investigators were not blinded to allocation during experiments and outcome assessment.

## Reporting summary

Further information on research design is available in the Nature Portfolio Reporting Summary linked to this article.

## Data availability

The sequencing data generated in this study have been deposited in the NCBI GEO database under accession code GSE235876. Source data are provided with this paper.

## Code availability

Custom scripts used in this study are available from GitHub: https://github.com/Henikoff/FFPE.

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

## Acknowledgements
We thank Christine Codomo, Doris Xu and Terri Bryson for technical assistance, Iris Luk for generating the cholangiocarcinoma model, Matthew Fitzgibbon for bioinformatics support, the Fred Hutch Genomics Shared Resource for sequencing and data processing and the Fred Hutch Experimental Histopathology Shared Resource for FFPE processing of liver samples. This work was supported by the Howard Hughes Medical Institute (S.H.) and grant # T32CA009515 from the National Cancer Institute (R.M.P.).

## Author contributions
S.H. and E.C.H. conceived the study; S.H. and R.M.P. performed the experiments; F.S., Z.R.R., S.K. and E.C.H. provided critical materials; D.H.J. advised on the methods; S.H. and J.G.H. analyzed the data; S.H. wrote the manuscript; S.H., J.G.H., K.A., S.K. and E.C.H. reviewed and edited the manuscript, and all authors approved the manuscript.

## Competing interests
S.H. is an inventor in a USPTO patent application filed by the Fred Hutchinson Cancer Center pertaining to CUTAC and FFPE-CUTAC (application number 63/505,964). The other authors declare no competing interests.
