## [Peer Review File · Nature Communications]

REVIEWER COMMENTS

Reviewer #1 (Remarks to the Author):

In this manuscript, Henikoff and colleagues describe a modified protocol for using CUT&TAG sequencing of formalin-fixed paraffin-embedded (FFPE) samples. They describe a simplified method, whereby methodological adaptations such as increased temperature, are sufficient to provide a simplified protocol with reduce time, and to produce high-resolution sequencing and mapping of RNA Pol II at enhancers and promoters in FFPE samples. The protocol produced much cleaner signal than previously described ones, such as FACT-seq. As proof of principle of their new methodology, the authors demonstrate that they are able to identify differences in the genomic landscape of different mouse brain tumor specimens, compared to normal tissues. The authors conclude that their new approach could be used to produce epigenetic profiles in archived biological samples, enabling new biological questions to be asked in historically collected samples. Despite the clear improvement in the technology, we believe the manuscript falls short in providing a significant advance, both technological and conceptually, to warrant publication in Nat. Comm., as described in detail below.

Major Comments:

1. I would like to first praise the authors on the adaptations that they have made to their existing CUT&TAG methodology. The data they provide is robust and supports the conclusions made in the manuscript, particularly pertaining to the improvements in sensitivity and specificity of their new approach.
2. In addition, the manuscript is extremely well-written and will be accessible to non-specialist readers. Given the technical nature of the manuscript, this is particularly well done.
3. Whilst the adaptations described in this manuscript clearly improve the existing methodologies, there are lingering concerns about the broader applications of this new approach, which is an important consideration for the readership of a non-specialist journal like Nature Communications. For example, the limited utility with H3K27Ac and RNAPol II antibodies, will limit the use somewhat of this approach. In this context, can the authors explain in more detail why histone marks associated with more open chromatin i.e., H3K36me3 and H3K4me3, did not work with this approach? Can they explore ways to improve the technology as to make it work for other chromatin marks/factors?
4. It is intriguing the major bias towards increased peaks in tumor samples compared to normal tissues, which is particularly evident for the H3K27Ac mark. Can the authors speculate why?

5. Whilst the new methodologies undoubtedly improve the sensitivity and specificity of the technique, the new approaches do represent a tweaking of existing approaches that have been well characterized and are now, much to the authors credit, routinely used in the field. For broader interest to the readership of Nature Communications, I think it would be necessary to expand on some of the biological findings that are alluded to in the manuscript. For instance, it was surprising that the authors show few clear examples of the top differential peaks in specific genes (the distinct pausing peak in PDGFB, major increase in all marks in Ccnd1, etc.) yet they did not provide information on the expression levels of these particular genes (instead, they show another group of genes where they correlated CUTAC with RNA-seq, but for genes where the CUTAC signal was much less clear). A follow up of these unique chromatin changes, how expression is affected, and the potential consequences of such expression in driving tumorigenesis, would provide strong support for the utility of the described protocol.

Reviewer #2 (Remarks to the Author):

Henikoff and team have introduced over the past years a number of methods, and variations thereof, which have transformed epigenomic profiling in cells, nuclei, low input and single cells. These methods have been developed to study post-translational modifications on histones and their genomic location. These are now standard methods in many labs over the world. In this study “Epigenomic analysis of Formalin-fixed paraffin-embedded samples by CUT&Tag”, they have adapted one of their techniques, Cut&Tag, to be used for difficult samples that have been preserved by fixation with formaldehyde. This fixation will have impact on epitope availability and other artifacts associate with crosslinking therefore conditions of the protocol need to be changed. This methodology is much desired by researchers having no alternative to FFPE samples in particular from patient samples.

The most relevant alternatives to the FFPE CUT&Tag is FACT-seq, which is a longer protocol and is associated with concerns over variability. In the report the authors demonstrate that their method is clearly superior on many parameters to FACT-seq. Interestingly, because of the relative low input of material needed, this method can distinguish cell identities, separating normal from tumour cells which could be used for clinical applications where perhaps other tumour markers are not available.

This is a very nice study, well executed and will be important for the scientific community, where sometimes no other sample than FFPE are available. Some of the findings are also quite interesting, for example reporting on the types bacteria that can contaminate FFPE samples.

Perhaps there is one major concern that I have since the authors wrote: “this protocol failed for H3K4me2 and H3K4me3” – how widely applicable is this method for histone modifications? Have they tried other modifications than H3K27ac and H3K4me2 and H3K4me3?

Questions:

1. It's not clear if there will be a different name to this protocol: is it FFPE Cut&Tag? Is it FFPE-CUTAC? Because there are modifications from the Cut&Tag protocol perhaps the title should feature a new name instead of just Cut&Tag?
2. The authors wrote: "We found that the Conca-navalin A (ConA) beads used for standard CUT&Tag bound sufficiently well to sheared FFPE fragments regardless of whether they had been prepared from samples deparaffinized using a xylene or a mineral oil procedure." Is this data shown?
3. In Figure 4 can they use the same scale for the FDR to make the comparisons easier?
4. Also in Figure 4, perhaps important to mention the input sample amount here not just equal nr of reads. ChIP-seq input must be very high compared to C&T and Fact-seq

Reviewer #1 (Remarks to the Author):

In this manuscript, Henikoff and colleagues describe a modified protocol for using CUT&TAG sequencing of formalin-fixed paraffin-embedded (FFPE) samples. They describe a simplified method, whereby methodological adaptations such as increased temperature, are sufficient to provide a simplified protocol with reduce time, and to produce high-resolution sequencing and mapping of RNA Pol II at enhancers and promoters in FFPE samples. The protocol produced much cleaner signal than previously described ones, such as FACT-seq. As proof of principle of their new methodology, the authors demonstrate that they are able to identify differences in the genomic landscape of different mouse brain tumor specimens, compared to normal tissues. The authors conclude that their new approach could be used to produce epigenetic profiles in archived biological samples, enabling new biological questions to be asked in historically collected samples. Despite the clear improvement in the technology, we believe the manuscript falls short in providing a significant advance, both technological and conceptually, to warrant publication in Nat. Comm., as described in detail below.

Reviewer 1's point is well-taken. We have significantly improved the manuscript technologically by omitting the deparaffinization step thus avoiding the use of organic reagents and adding an additional even simpler protocol in which all of the steps through tagmentation are performed on the FFPE slide itself (new Figure 2a). Our on-slide protocol has allowed us to dissect and compare tumor versus normal sections from the same 10-micron brain specimen, which confirms that tumor RNAPII is inherently more accessible in FFPEs than normal (new Figure 2e-f). We had not felt that making this point was justified in the original submission as it was based on comparisons between tumor and normal from different brain specimens and was not seen in liver tumor-versus-normal comparisons. This additional technological advance also has led to an important conceptual advance, which we think addresses the shortcomings of the original submission. As described below in response to the Reviewer's Point 5, we have also used the new dissection data to discover and precisely map Mir124a-1hg, Mir124a-2hg and Mir670 microRNA genes as the most RNAPII down-regulated of all genes by far in RELA fusion-driven tumor FFPEs (new Figure 6c-d). In contrast, these genes are far down RNA-seq list ranked by false discovery rate, as Mir124a-1hg ranks 9,913, Mir124a-2hg ranks 6,045 and Mir670 ranks 21,262 (Supplementary Table 4). As the Mir124a microRNA is an abundant neuronal differentiation factor, a tumor suppressor, and a unique human biomarker for *Helicobacter pylori* infection and gastric cancer risk, our exciting discovery demonstrates that FFPE-CUTAC can make discoveries that high-quality RNA-seq data had completely missed while providing unexpected new insights for cancer biology. Because of the profound advantage of FFPE-CUTAC over RNA-seq in identifying and mapping important microRNAs, we now refer to this unique ability of FFPE-CUTAC in the Abstract and Discussion (lines 435-444).

Major Comments:

1. I would like to first praise the authors on the adaptations that they have made to their existing CUT&TAG methodology. The data they provide is robust and supports the conclusions made in the manuscript, particularly pertaining to the improvements in sensitivity and specificity of their new approach.

We thank Reviewer 1 for this overall positive assessment of our method.

2. In addition, the manuscript is extremely well-written and will be accessible to non-specialist readers. Given the technical nature of the manuscript, this is particularly well done.

We thank Reviewer 1 for appreciation of our efforts to make the manuscript accessible to the non-specialist, and as outlined in response to both reviewers, the addition of biological interpretations that were missing from the original submission should make the manuscript more suitable for a general audience.

3. Whilst the adaptations described in this manuscript clearly improve the existing methodologies, there are lingering concerns about the broader applications of this new approach, which is an important consideration for the readership of a non-specialist journal like Nature Communications. For example, the limited utility with H3K27Ac and RNAPol II antibodies, will limit the use somewhat of this approach.

Whereas the three antibodies (RNAPII-Ser5, RNAPII-Ser2,5 and H3K27ac) are limited relative to all available antibodies, they are the ones most informative for regulatory element identification. Moreover, the use of antibodies for paused RNAPII provides ground-truth validation for open chromatin, a concept that dates back to 1990. That is, until we showed with CUTAC that sites of open chromatin were indeed sites of paused RNAPII marked by RNAPII-Ser5, different assays for open chromatin, such as DNaseI hypersensitivity mapping and ATAC-seq that typically differ in subtle or not-so-subtle ways had no ground truth to decide between them. We now make this point in the text (lines 420-425).

In this context, can the authors explain in more detail why histone marks associated with more open chromatin i.e., H3K36me3 and H3K4me3, did not work with this approach? Can they explore ways to improve the technology as to make it work for other chromatin marks/factors?

We have added H3K27me3 FFPE data in Supplementary Figure 2 which indicates that there is only weak signal over broad Polycomb domains that are clearly distinct in published CUT&Tag data. As H3K27me3 antibodies are used as CUT&Tag controls because of their high abundance and outstanding signal-to-noise, we might account for the complete failure of other antibodies as a result of their much less abundant epitopes (lines 174-177).

4. It is intriguing the major bias towards increased peaks in tumor samples compared to normal tissues, which is particularly evident for the H3K27Ac mark. Can the authors speculate why?

In the original submission we pointed to the observation of hypertranscription in aggressive human cancers as a possible explanation, and with the addition of tumor-versus-normal data from the same brain section we now confirm this excess of up-regulation in a more rigorous comparison. We have modified the Discussion to strengthen this speculation (lines 449-455).

5. Whilst the new methodologies undoubtedly improve the sensitivity and specificity of the technique, the new approaches do represent a tweaking of existing approaches that have been well characterized and are now, much to the authors credit, routinely used in the field. For broader interest to the readership of Nature Communications, I think it would be necessary to expand on some of the biological findings that are alluded to in the manuscript. For instance, it

was surprising that the authors show few clear examples of the top differential peaks in specific genes (the distinct pausing peak in PDGFB, major increase in all marks in Ccnd1, etc.) yet they did not provide information on the expression levels of these particular genes (instead, they show another group of genes where they correlated CUTAC with RNA-seq, but for genes where the CUTAC signal was much less clear). A follow up of these unique chromatin changes, how expression is affected, and the potential consequences of such expression in driving tumorigenesis, would provide strong support for the utility of the described protocol.

We thank Reviewer 1 for this suggestion. Indeed, we had provided little direct evidence for the utility of our method beyond identifying biomarkers validated by RNA-seq. Specifically, we chose genes based on our previous YAP1 RNA-seq data to provide independent confirmation of FFPE-CUTAC tumor-versus-normal comparison using an orthogonal modality. The advantage of using well-studied fusion transgene mouse models in our study is that we already know how they drive cancer, so further investigation into the biology of these tumors will not further demonstrate the utility of FFPE-CUTAC. However, we can expand on some of the biological findings by asking whether FFPE-CUTAC can identify functional targets that RNA-seq cannot. Accordingly, we now show that the top down-regulated locus in a comparison between RELA and normal tissue dissected from the same FFPE is Mir124a-hg1 (new Figure 6c), a microRNA locus, which is Rank 9,913 of 23,551 mouse genes by FDR (Supplementary Table 4). Indeed, the top 10 down-regulated cCREs are either Mir124a-hg1 or Mir124a-hg2 and these together with the next best down-regulated hit, Mir670, account for 15 of the top 25 down-regulated cCREs (Supplementary Table 3). In humans the Mir124a gene is a unique DNA methylation marker locus for *Helicobacter pylori* infection, which correlates with methylation of gastric cancer drivers. Moreover, the entire locus is embedded in a cluster of 27 cCREs, and all of the replicates show a broad RNAPII signal in normal tissue but not RELA-driven tumor encompassing the entire cluster. RNA-seq can only show levels of expression, whereas the value of FFPE-CUTAC is both in identifying biomarkers and mapping them to the corresponding regulatory elements in the genome. We now make this point in the Discussion.

Reviewer #2 (Remarks to the Author):

Henikoff and team have introduced over the past years a number of methods, and variations thereof, which have transformed epigenomic profiling in cells, nuclei, low input and single cells. These methods have been developed to study post-translational modifications on histones and their genomic location. These are now standard methods in many labs over the world. In this study “Epigenomic analysis of Formalin-fixed paraffin-embedded samples by CUT&Tag”, they have adapted one of their techniques, Cut&Tag, to be used for difficult samples that have been preserved by fixation with formaldehyde. This fixation will have impact on epitope availability and other artifacts associate with crosslinking therefore conditions of the protocol need to be changed. This methodology is much desired by researchers having no alternative to FFPE samples in particular from patient samples.

The most relevant alternatives to the FFPE CUT&Tag is FACT-seq, which is a longer protocol and is associated with concerns over variability. In the report the authors demonstrate that their method is clearly superior on many parameters to FACT-seq. Interestingly, because of the relative low input of material needed, this method can distinguish cell identities, separating

normal from tumour cells which could be used for clinical applications where perhaps other tumour markers are not available.

This is a very nice study, well executed and will be important for the scientific community, where sometimes no other sample than FFPE are available. Some of the findings are also quite interesting, for example reporting on the types bacteria that can contaminate FFPE samples.

Perhaps there is one major concern that I have since the authors wrote: “this protocol failed for H3K4me2 and H3K4me3” – how widely applicable is this method for histone modifications?

Have they tried other modifications than H3K27ac and H3K4me2 and H3K4me3?

Reviewer 1 had the same question, and we have added H3K27me3 FFPE data in the new Supplementary Figure 2 showing that there is significant but weak signal over broad Polycomb domains, so that we might account for the failure of antibodies as a result of their much less abundant epitopes.

Questions:

1. It's not clear if there will be a different name to this protocol: is it FFPE Cut&Tag? Is it FFPE-CUTAC? Because there are modifications from the Cut&Tag protocol perhaps the title should feature a new name instead of just Cut&Tag?

We call it FFPE-CUTAC throughout the paper to distinguish it from standard CUTAC. FFPE-CUTAC now uses two very different workflows, one like standard CUT&Tag and the other on-slide as outlined in the new Figure 2a. Since CUTAC uses the same workflow as standard CUT&Tag, only using a low ionic strength buffer for tagmentation, and CUT&Tag is familiar to users whereas CUTAC is not, and FFPE-CUTAC will not be properly understood at all in a title, we prefer to use CUT&Tag in the title and spell out “FFPE”.

2. The authors wrote: “We found that the Concanavalin A (ConA) beads used for standard CUT&Tag bound sufficiently well to sheared FFPE fragments regardless of whether they had been prepared from samples deparaffinized using a xylene or a mineral oil procedure.” Is this data shown?

We have found that our 80-90°C treatment suffices to melt and float off the paraffin, confirming reports that hot water treatment obviates the need for deparaffinization using organic solvents such as xylene or mineral oil. Accordingly, we have modified the cartoon in the new Figure 2a showing the workflow beginning with 85°C incubation and have shortened this sentence to: “We found that the Concanavalin A (ConA) beads used for standard CUT&Tag, bound sufficiently well to sheared FFPE fragments.”

3. In Figure 4 can they use the same scale for the FDR to make the comparisons easier?

Agreed. All volcano plots in the same figure (Figures 4, 6 and 7 and Supplementary Figure 4) are now on the same scale to make comparisons easier.

4. Also in Figure 4, perhaps important to mention the input sample amount here not just equal nr of reads. ChIP-seq input must be very high compared to C&T and Fact-seq

We agree and now provide that information in the legend.

REVIEWERS' COMMENTS

Reviewer #1 (Remarks to the Author):

In this revised version, Henikoff and colleagues tried to address the reviewers' concerns. Although the responses were for the most part text changes, the additional clarity that the new discussions provide did strengthen the manuscript. They also expanded the experiments to assess the biological significance of the specific gene targets they found (particularly Mir124a in gastric cancer). All in all, I think the manuscript is improved, and, even if I am still hesitant on how much novelty the described protocol brings compared to their previous publications, it is clear the protocol could immensely expand the analysis of FFPE blocks, and thus I think it will be a great resource.

Reviewer #2 (Remarks to the Author):

The authors have addressed my points. Despite technical limitations in using this method to map histone methylation marks, caused by formalin fixation, there is a clear and focused applicability of FFPE-CUTAC to identify regulatory sites in a cost effective manner, using low input material, in a streamlined sample preparation. This will be attractive to many scientists, not limited to clinicians, clinical studies and those having access to large retrospective clinical cohorts. By mapping paused RNA PolIII, the authors came across a significant finding whereby lowly expressing micro RNAs can be identified, which would have been missed by RNA-seq. This will likely fuel many future discoveries in cancer biology.

Reviewer #1 (Remarks to the Author):

In this revised version, Henikoff and colleagues tried to address the reviewers' concerns. Although the responses were for the most part text changes, the additional clarity that the new discussions provide did strengthen the manuscript. They also expanded the experiments to assess the biological significance of the specific gene targets they found (particularly Mir124a in gastric cancer). All in all, I think the manuscript is improved, and, even if I am still hesitant on how much novelty the described protocol brings compared to their previous publications, it is clear the protocol could immensely expand the analysis of FFPE blocks, and thus I think it will be a great resource.

We thank Reviewer #1 for their enthusiasm for the method. In the final revised version we have strengthened the case for novelty compared to previous work by explaining the logic for identifying regulatory elements by FFPE-CUTAC for biomarker discovery. Specifically, RNA-seq cannot be used to *discover* biomarkers, only *confirm* them as being up- or down-regulated. Here is revised text in the second paragraph of the Discussion to clarify that point: "Changes in TF mRNA abundance with cell type changes are swamped out by changes in more abundant mRNAs. mRNA abundance is also affected by complex regulatory mechanisms occurring during and after transcriptional elongation, including multiple modes of processing and export to the cytoplasm, resulting in undifferentiated volcano plots that resemble erupting volcanos (**Supplementary Figure S4**). Our comparisons were against RNA-seq data from fresh or frozen tissue, whereas RNA-seq results from FFPEs are much poorer owing to serious degradation and off-target reads (58). In contrast, paused RNAPII is a critical checkpoint for transcriptional activation, and so its abundance at a regulatory element is a direct measure of transcriptional competence, and cancer driver and tumor suppressor loci stand out (**Figures 5 and 6c-d**)."

Reviewer #2 (Remarks to the Author):

The authors have addressed my points. Despite technical limitations in using this method to map histone methylation marks, caused by formalin fixation, there is a clear and focused applicability of FFPE-CUTAC to identify regulatory sites in a cost effective manner, using low input material, in a streamlined sample preparation. This will be attractive to many scientists, not limited to clinicians, clinical studies and those having access to large retrospective clinical cohorts. By mapping paused RNA PolII, the authors came across a significant finding whereby lowly expressing micro RNAs can be identified, which would have been missed by RNA-seq. This will likely fuel many future discoveries in cancer biology.

We thank Reviewer #1 for their enthusiasm for the prospects of our method for future investigations beyond the clinic.